# RelaxLoss: Defending Membership Inference Attacks without Losing Utility

**Dingfan Chen**[1]    **Ning Yu**[2,3,4*]    **Mario Fritz**[1]
[1]CISPA Helmholtz Center for Information Security    [2]Salesforce Research
[3]University of Maryland    [4]Max Planck Institute for Informatics
{dingfan.chen, fritz}@cispa.de   ning.yu@salesforce.com

## Abstract

As a long-term threat to the privacy of training data, membership inference attacks (MIAs) emerge ubiquitously in machine learning models. Existing works evidence strong connection between the distinguishability of the training and testing loss distributions and the model's vulnerability to MIAs. Motivated by existing results, we propose a novel training framework based on a *relaxed loss* (**RelaxLoss**) with a more achievable learning target, which leads to narrowed generalization gap and reduced privacy leakage. RelaxLoss is applicable to any classification model with added benefits of easy implementation and negligible overhead. Through extensive evaluations on five datasets with diverse modalities (images, medical data, transaction records), our approach consistently outperforms state-of-the-art defense mechanisms in terms of resilience against MIAs as well as model utility. Our defense is the first that can withstand a wide range of attacks while preserving (or even improving) the target model's utility. Source code is available at https://github.com/DingfanChen/RelaxLoss.

## 1 Introduction

While deep learning (DL) models have achieved tremendous success in the past few years, their deployments in many sensitive domains (e.g., medical, financial) bring privacy concerns since data misuse in these domains induces severe privacy risks to individuals. In particular, modern deep neural networks (NN) are prone to memorize training data due to their high capacity, making them vulnerable to privacy attacks that extract detailed information about the individuals from models (Shokri et al., 2017; Song et al., 2017; Yeom et al., 2018) .

In membership inference attack (MIA), an adversary attempts to identify whether a specific data sample was used to train a target victim model. This threat is pervasive in various data domains (e.g., images, medical data, transaction records) and inevitably poses serious privacy threats to individuals (Shokri et al., 2017; Nasr et al., 2018; Salem et al., 2019), even given only black-box access (query inputs in, posterior predictions out) (Shokri et al., 2017; Salem et al., 2019; Song & Mittal, 2020) or partially observed output predictions (e.g., top-k predicted labels) (Choo et al., 2020).

Significant advances have been achieved to defend against MIAs. Conventionally, regularization methods designed for mitigating overfitting such as dropout (Srivastava et al., 2014) and weight-decay (Geman et al., 1992) are regarded as defense mechanisms (Salem et al., 2019; Jia et al., 2019; Shokri et al., 2017). However, as conveyed by Kaya et al. (2020); Kaya & Dumitras (2021), vanilla regularization techniques (which are not designed for MIA), despite slight improvement towards reducing the generalization gap, are generally unable to eliminate MIA. In contrast, recent works design defenses tailored to MIA. A common strategy among such defenses is adversarial training (Goodfellow et al., 2014b;a), where a surrogate attack model (represented as a NN) is used to approximate the real attack and subsequently the target model is modified to maximize prediction errors of the surrogate attacker via adversarial training. This strategy contributes to remarkable success in defending NN-based attacks (Nasr et al., 2018; Jia et al., 2019). However, these methods are greatly

---

[*]This work was done when Ning Yu was in a joint Ph.D. program with the University of Maryland and Max Planck Institute for Informatics.

restricted by strong assumptions on attack models, thereby failing to generalize to novel attacks unanticipated by the defender (e.g., a simple metric-based attack) (Song & Mittal, 2020). In order to defend attacks beyond the surrogate one, differentially private (DP) training techniques (Abadi et al., 2016; Papernot et al., 2016; 2018) that provide strict guarantees against MIA are exploited. Nevertheless, as evidenced by Rahman et al. (2018); Jia et al. (2019); Hayes et al. (2019); Jayaraman & Evans (2019); Chen et al. (2020); Kaya & Dumitras (2021), incorporating DP constraints inevitably compromises model utility and increases computation cost.

In this paper, we present an effective defense against MIAs while avoiding negative impacts on the defender's model utility. Our approach is built on two main insights: *(i)* the optimal attack only depends on the sample loss under mild assumptions of the model parameters (Sablayrolles et al., 2019); *(ii)* a large difference between the training loss and the testing loss provably causes high membership privacy risks (Yeom et al., 2018). By intentionally 'relaxing' the target training loss to a level which is more achievable for the test loss, our approach narrows the loss gap and reduces the distinguishability between the training and testing loss distributions, effectively preventing various types of attacks in practice. Moreover, our approach allows for a utility-preserving (or even improving) defense, greatly improving upon previous results. As a practical benefit, our approach is easy to implement and can be integrated into any classification models with minimal overhead.

**Contributions.** *(i)* We propose **RelaxLoss**, a simple yet effective defense mechanism to strengthen a target model's resilience against MIAs without degrading its utility. To the best of our knowledge, our approach for the first time addresses a wide range of attacks while preserving (or even improving) the model utility. *(ii)* We derive our method from a Bayesian optimal attacker and provide both empirical and analytical evidence supporting the main principles of our approach. *(iii)* Extensive evaluations on five datasets with diverse modalities demonstrate that our method outperforms state-of-the-art approaches by a large margin in membership inference protection and privacy-utility trade-off.

## 2 RELATED WORK

**Membership Inference Attack.** Inferring membership information from deep NNs has been investigated in various application scenarios, ranging from the white-box setting where the whole target model is released (Nasr et al., 2019; Rezaei & Liu, 2020) to the black-box setting where the complete/partial output predictions are accessible to the adversary (Shokri et al., 2017; Salem et al., 2019; Yeom et al., 2018; Sablayrolles et al., 2019; Song & Mittal, 2020; Choo et al., 2020; Hui et al., 2021; Truex et al., 2019). An adversary first determines the most informative features (depending on the application scenarios) that faithfully reflect the sample membership (e.g., logits/posterior predictions (Shokri et al., 2017; Salem et al., 2019; Jia et al., 2019), loss values (Yeom et al., 2018; Sablayrolles et al., 2019), and gradient norms (Nasr et al., 2019; Rezaei & Liu, 2020)), and subsequently extracts common patterns in these features among the training samples for identifying membership. In this work, we work towards an effective defense by suppressing the common patterns that an optimal attack relies on.

**Defense.** Existing defense mechanisms against MIA are mainly divided into three main categories: *(i)* regularization techniques to alleviate model overfitting, *(ii)* adversarial training to confuse surrogate attackers, and *(iii)* a differentially private mechanism offering rigorous privacy guarantees. Our proposed approach can be regarded as a regularization technique owing to its effect in reducing generalization gap. Unlike previous regularization techniques, our method is explicitly tailored towards defending MIAs by reducing the information that an attacker can exploit, leading to significantly better defense effectiveness. Algorithmically, our approach shares similarity with techniques that suppress the target model's confidence score predictions (e.g., label-smoothing (Guo et al., 2017; Müller et al., 2019) and confidence-penalty (Pereyra et al., 2017)), but ours is fundamentally different in the sense that we modulate the loss distribution with gradient ascent.

Previous state-of-the-art defense mechanisms against MIA, such as Memguard (Jia et al., 2019) and Adversarial Regularization (Nasr et al., 2018), are built on top of the idea of adversarial training (Goodfellow et al., 2014b;a). Such approaches usually rely on strong assumptions about attack models, making their effectiveness highly dependent on the similarity between the surrogate and the real attacker (Song & Mittal, 2020). In contrast, our method does not rely on any assumptions about the attack model, and has shown consistent effectiveness across different attacker types.

Differential privacy (Dwork, 2008; Dwork et al., 2014; Abadi et al., 2016; Papernot et al., 2016) provides strict worst-case guarantees against arbitrarily powerful attackers that exceed practical limits, but inevitably sacrifices model utility (Rahman et al., 2018; Jia et al., 2019; Hayes et al., 2019; Chen et al., 2020; Kaya & Dumitras, 2021; Jayaraman & Evans, 2019) and meanwhile increases computation burden (Goodfellow, 2015; Dangel et al., 2019). In contrast, we focus on practically realizable attacks for utility-preserving and computationally efficient defense.

## 3 PRELIMINARIES

**Notations.** We denote by $z_i = (x_i, y_i)$ one data sample, where $x_i$ and $y_i$ are the feature and the one-hot label vector, respectively. $f(\cdot; \theta)$ represents a classification model parametrized by $\theta$, and $p = f(x; \theta) \in [0, 1]^C$ denotes the predicted posterior scores (after the final softmax layer) where $C$ denotes the number of classes. $\mathbb{1}$ denotes the indicator function, i.e., $\mathbb{1}[p]$ equals 1 if the predicate $p$ is true, else 0. We use subscripts for sample index and superscripts for class index.

**Attacker's Assumptions.** We consider the standard setting of MIA: the attacker has access to a query set $\mathbb{S} = \{(z_i, m_i)\}_{i=1}^N$ containing both member (training) and non-member (testing) samples drawn from the same data distribution $P_{\text{data}}$, where $m_i$ is the membership attribute ($m_i = 1$ if $z_i$ is a member). The task is to infer the value of the membership attribute $m_i$ associated with each query sample $z_i$. We design defense for a general attack with full access to the target model. The attack $\mathcal{A}(z_i, f(\cdot; \theta))$ is a binary classifier which predicts $m_i$ for a given query sample $z_i$ and a target model parametrized by $\theta$. The Bayes optimal attack $\mathcal{A}_{opt}(z_i, f(\cdot; \theta))$ will output 1 if the query sample is more likely to be contained in the training set, based on the real underlying membership probability $P(m_i = 1 | z_i, \theta)$, which is usually formulated as a non-negative log ratio:

$$\mathcal{A}_{opt}(z_i, f(\cdot; \theta)) = \mathbb{1}\left[\log \frac{P(m_i = 1 | z_i, \theta)}{P(m_i = 0 | z_i, \theta)} \geq 0\right] \tag{1}$$

**Defender's Assumptions.** We closely mimic an assumption-free scenario in designing our defense method. In particular, we consider a knowledge-limited defender which: *(i)* does not have access to additional public (unlabelled) training data (in contrast to Papernot et al. (2016; 2018)); and *(ii)* lacks prior knowledge of the attack strategy (in contrast to Jia et al. (2019); Nasr et al. (2018)). For added rigor, we also study attacker's countermeasures to our defense in Section 6.4.

## 4 RELAXLOSS

The ultimate goal of the defender is two-fold: *(i) privacy:* reducing distinguishability of member and non-member samples; *(ii) utility:* avoiding the sacrifice of the target model's performance. We hereby introduce each component of our method targeting at privacy (Section 4.1) and utility (Section 4.2).

### 4.1 PRIVACY: REDUCE DISTINGUISHABILITY VIA RELAXED TARGET LOSS

We begin by exploiting the dependence of attack success rate on the sample loss (Yeom et al., 2018; Sablayrolles et al., 2019). In particular, a large gap in the expected loss values on the member and non-member data, i.e., $\mathbb{E}[\ell]_{\text{non}} - \mathbb{E}[\ell]_{\text{mem}}$, is proved to be sufficient for conducting attacks (Yeom et al., 2018). Along this line, Sablayrolles et al. (2019) further show that the Bayes optimal attack only depends on the sample loss under a mild posterior assumption of the model parameter: $\mathcal{P}(\theta | z_1, .., z_n) \propto e^{-\frac{1}{T} \sum_{i=1}^N m_i \cdot \ell(\theta, z_i)}$[1]. Formally,

$$\mathcal{A}_{opt}(z_i, f(\cdot; \theta)) = \mathbb{1}[-\ell(\theta, z_i) > \tau(z_i)] \tag{2}$$

where $\tau$ denotes a threshold function [2]. Intuitively, Equation 2 shows that $z_i$ is likely to be used for training if the target model exhibits small loss value on it. These results motivate our approach to mitigate MIAs by reducing distinguishablitiy between the member and non-member loss distributions.

---

[1] This corresponds to a Bayesian perspective, i.e., $\theta$ is regarded as a random variable that minimizes the empirical risk $\sum_{i=1}^N m_i \cdot \ell(\theta, z_i)$. $T$ is the temperature that captures the stochasticity.

[2] We summarize both the strategy MALT and MAST from Sablayrolles et al. (2019) in Equation 2, where $\tau$ is a constant function for MALT.

**Relaxing Loss Target with Gradient Ascent.**
Directly operating on member and non-member loss distributions, however, is impractical, since the exact distributions are intractable and a large amount of additional hold-out samples are required for estimating the distribution of non-member data. In order to bypass these issues and reduce the distinguishability between the member and non-member loss distributions, we propose to simplify the problem by considering the mean of the loss distributions, and subsequently set a *more achievable* mean value for the target loss, where the loss target is relaxed to a level that is easier to be achieved for the non-member data.

Algorithmically, instead of pursuing zero training loss of the target model, we relax the target mean loss value $\alpha$ to be larger than zero and apply a *gradient ascent* step as long as the average loss of the current batch is smaller than $\alpha$.

## 4.2 UTILITY: APPLY POSTERIOR FLATTENING AND NORMAL GRADIENT OPERATIONS

With the relaxed target loss, the predicted posterior score of the ground-truth class $p^{gt}$ is no longer maximized towards 1. If the probability mass of all non-ground-truth classes $1 - p^{gt}$ concentrates on only few of them (e.g., hard samples that are close to the decision boundary between two classes), it is very likely that one non-ground-truth class has a score larger than $p^{gt}$ (i.e., $\max_{c,c \neq gt} p^c > p^{gt}$), thus leading to incorrect predictions. To address this issue, we propose to encourage a large margin between the prediction score of the ground-truth-class and the others by *flattening the target posterior* scores for non-ground-truth classes. Specifically, we dynamically construct softlabels during each epoch by: *(i)* retaining the score of the ground-truth class, i.e., the current predicted value $p^{gt}$, and *(ii)* re-allocating the remaining probability mass evenly to all non-ground-truth classes.

---

**Algorithm 1:** RelaxLoss

**Input:** Dataset $\{(\boldsymbol{x}_i, \boldsymbol{y}_i)\}_{i=1}^N$, training epochs $E$, learning rates $\tau$, batch size $B$, number of output classes $C$, target loss value $\alpha$

**Output:** Model $f(\cdot; \boldsymbol{\theta})$ with parameters $\boldsymbol{\theta}$

Initialize model parameter $\boldsymbol{\theta}$ ;

**for** *epoch* **in** $\{1, ..., E\}$ **do**
    **for** *batch_index* **in** $\{1, ..., K\}$ **do**
        Get sample batch $\{(\boldsymbol{x}_i, \boldsymbol{y}_i)\}_{i=1}^B$
        Perform forward pass: $\boldsymbol{p}_i = f(\boldsymbol{x}_i; \boldsymbol{\theta})$
        Compute cross entropy loss $\mathcal{L}(\boldsymbol{\theta})$ on the batch
        **if** $\mathcal{L}(\boldsymbol{\theta}) \geq \alpha$ **then**
            // gradient descent
            $\boldsymbol{\theta} \leftarrow \boldsymbol{\theta} - \tau \cdot \nabla \mathcal{L}(\boldsymbol{\theta})$
        **else**
            **if** *epoch* $\%2 = 0$ **then**
                // gradient ascent
                $\boldsymbol{\theta} \leftarrow \boldsymbol{\theta} + \tau \cdot \nabla \mathcal{L}(\boldsymbol{\theta})$
            **else**
                // posterior flattening
                Construct softlabel $\boldsymbol{t}_i$ with

$$t_i^c = \begin{cases} p_i^c & \text{if } y_i^c = 1 \\ (1 - p_i^c)/(C - 1) & \text{otherwise} \end{cases}$$

                Compute cross entropy loss with the softlabel: [a])
$$\ell(\boldsymbol{\theta}, \boldsymbol{z}_i) = -\sum_{c=1}^C \text{sg}[t_i^c] \log p_i^c$$
$$\mathcal{L}(\boldsymbol{\theta}) = \frac{1}{B} \sum_{i=1}^B \ell(\boldsymbol{\theta}, \boldsymbol{z}_i)$$
                Update model parameters:
$$\boldsymbol{\theta} \leftarrow \boldsymbol{\theta} - \tau \nabla \mathcal{L}(\boldsymbol{\theta})$$
            **end**
        **end**
    **end**
**end**
**return** model $f(\cdot; \boldsymbol{\theta})$

---

[a]   sg stands for the stopgradient operator that is defined as identity at forward pass and has zero partial derivatives, i.e., $\boldsymbol{t}_i$ is a non-updated constant.

In summary, we run a repetitive training strategy to balance privacy and utility, which consists of two steps: *(i)* if the model is not well-trained, i.e., the current loss is larger than the target mean value $\alpha$, we run a normal gradient descent step; *(ii)* otherwise, we apply gradient ascent or the posterior flattening step (See Algorithm 1).

## 5 ANALYTICAL INSIGHTS

In this section, we analyze the key properties that explain the effectiveness of RelaxLoss. We provide both analytical and empirical evidence showing that RelaxLoss can *(i)* reduce the generalization gap, and *(ii)* increase the variance of the training loss distribution, both contributing to mitigating MIAs.

**RelaxLoss reduce the generalization gap.** We apply RelaxLoss to CIFAR-10 dataset and plot the resulting loss histograms in Figure 1. With a more achievable learning target, RelaxLoss blurs the

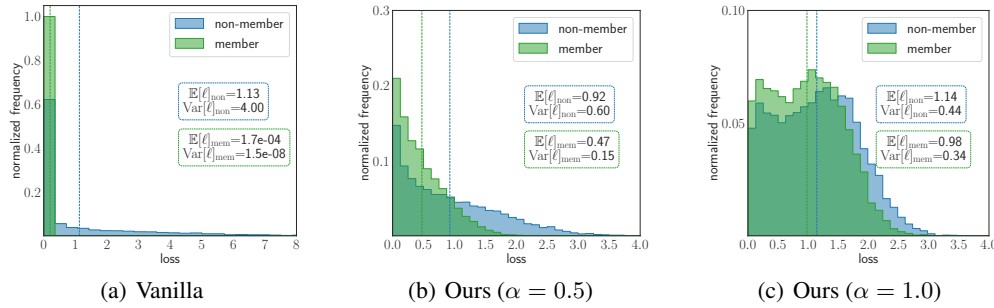

(a) Vanilla    (b) Ours ($\alpha = 0.5$)    (c) Ours ($\alpha = 1.0$)

**Figure 1:** Loss histograms on CIFAR-10 with ResNet20 architecture when applying (a) vanilla training, (b) our method with $\alpha = 0.5$, and (c) our method with $\alpha = 1.0$. The empirical mean and variance of the loss distributions are shown in the figure. The AUC of a loss thresholding attack equals to 0.84 in (a), 0.67 in (b), and 0.57 in (c). We observe that our method fits the target mean (Section 4.1), increases the variance of the training loss distribution and reduces the distinguishability between member and non-member distributions (Section 5).

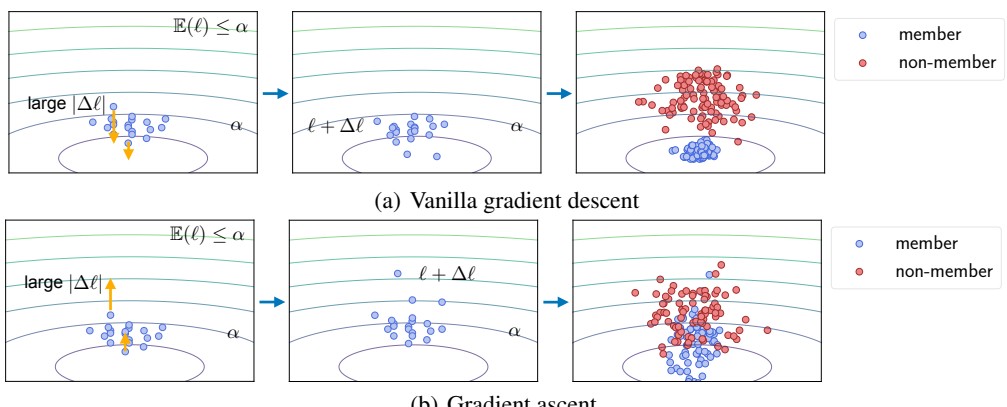

(a) Vanilla gradient descent

(b) Gradient ascent

**Figure 2:** Comparison between vanilla gradient descent and the gradient ascent step in RelaxLoss (demonstrated in 2D). The loss contour lines are plotted in the figure, the *bottom* part of which corresponds to a *low* loss region. The target loss level $\alpha$ is visualized in the figure. Training with vanilla gradient descent step results in near zero loss for member samples, and large loss values for non-member samples. In contrast, a large loss value $\ell$ tends to trigger large update $|\Delta \ell|$ during the gradient ascent step. As a result, RelaxLoss spreads out the training loss distribution and blurs the gap between the distributions (Section 5).

gap between the member and non-member loss distributions (Figure 1), which naturally leads to a narrowed generalization gap (Appendix Figure 7) and reduced privacy leakage (Yeom et al., 2018).

**RelaxLoss increases the variance of the training loss distribution.** We observe that RelaxLoss spreads out the training loss distribution (i.e., increase the variance) due to its gradient ascent step (Appendix A.1): large loss samples tend to have a more significant increase in its loss value during the gradient ascent step (See Figure 2 for demonstration). In contrast, except DP-SGD, existing defense methods do not have this property (Appendix C.10). Intuition suggests that the increase of training loss variance suppresses the common pattern among training losses and reduces the information that can be exploited by an attacker, thus contributing to the protection against attacks. To verify the association between the variance increasing effect and the defense effectiveness, we conduct experiments on CIFAR-10 dataset and measure the Pearson's correlation coefficients between the training loss variance and the attack AUC (Appendix Figure 6). With an overall score of -0.85 (-0.77 and -0.94 for black-box and white-box attacks, respectively), we conclude a fairly strong negative relationship. When considering a typical Gaussian assumption of the loss distributions (Yeom et al., 2018; Li et al., 2020), we further show this variance increasing property helps to lower an upper bound of the attack AUC, and provide formal analysis in Appendix A.2.

## 6 EXPERIMENTS

In this section, we rigorously evaluate the effectiveness of our defense across a wide range of datasets with diverse modalities, various strong attack models, and eight defense baselines representing the previous state of the art.

### 6.1 EXPERIMENTAL SETUP

**Settings.** We set up seven target models, trained on five datasets (CIFAR-10, CIFAR-100, CH-MNIST, Texas100, Purchase100) with diverse modalities (natural and medical images, medical records, and shopping histories). For image datasets, we adopt a 20-layer ResNet (He et al., 2016) and an 11-layer VGG (Simonyan & Zisserman, 2015); and for non-image datasets, we adopt MLPs with the same architecture and same training protocol as in previous works (Nasr et al., 2018; Jia et al., 2019). We evenly split each dataset into five folds and use each fold as the training/testing set for the target/shadow model[3], and use the last fold for training the surrogate attack model (for Jia et al. (2019); Shokri et al. (2017)). We fix the random seed and training setting for a fair comparison among different defense methods. See Appendix B for implementation details.

**Attack methods.** To broadly handle attack methods in our defense evaluation, we consider attacks in a variety of application scenarios (*black-box* and *white-box*) and strategies. We consider the following state-of-the-art attacks from two categories:*(i) White-box attacks*: Both Nasr et al. (2019); Rezaei & Liu (2020) are based on gradient norm thresholding. We denote these attacks by the type of gradient followed by its norm. **Grad-x** and **Grad-w** stand for the gradient w.r.t. the *input* and the *model parameters*, respectively; *(ii) Black-box attacks*: Salem et al. (2019) (denoted as **NN**, standing for the proposed neural network attack model). We adopt the implementation provided by Jia et al. (2019) and use the complete logits prediction as input to the attacker. Sablayrolles et al. (2019) (denoted as **Loss** for their loss thresholding method). We use a general threshold independent of the query sample, as the adaptive thresholding version is more expensive computational-wise with no or only marginal improvements. Song & Mittal (2020) (denoted as **Entropy** and **M-Entropy** for their proposed attack by thresholding the prediction entropy and a modified version, respectively.). We exclude attacks that only use partial output predictions (e.g., top-1 predicted label) from our evaluation as they are strictly weaker than the attacks we include above (Choo et al., 2020).

**Evaluation metrics.** We evaluate along two fronts: utility (measured by **test accuracy** of the victim model) and privacy. For privacy, in line with previous works (Song & Mittal, 2020; Jia et al., 2019; Sablayrolles et al., 2019; Shokri et al., 2017; Nasr et al., 2018; Salem et al., 2019), we consider the following two metrics: *(i)* attack **accuracy**: We evaluate the attack accuracy on a balanced query set, where a random guessing baseline corresponds to 50% accuracy. For threshold-based attack methods, following Song & Mittal (2020), we select the threshold value to be the one with the best attack accuracy on the shadow model and shadow dataset; *(ii)* attack **AUC**: The area under the receiver operating characteristic curve (AUC), corresponding to an integral over all possible threshold values, represents the degree of separability. A perfect defense mechanism corresponds to AUC=0.5.

### 6.2 COMPARISON TO BASELINES

**Defense Baselines.** We consider two state-of-the-art defense methods: **Memguard** (Jia et al., 2019) and Adversarial regularization (**Adv-Reg**) (Nasr et al., 2018). Additionally, we compare to five regularization methods: **Early-stopping**, **Dropout** (Srivastava et al., 2014), **Label-smoothing** (Guo et al., 2017; Müller et al., 2019), **Confidence-penalty** (Pereyra et al., 2017) and (Self-)**Distillation** (Hinton et al., 2015; Zhang et al., 2019). Moreover, we compare to differential private mechanism, i.e., Differentially private stochastic gradient descent (**DP-SGD**) (Abadi et al., 2016)[4]. We exclude defenses that additionally require public (unlabelled) data (Papernot et al., 2016; 2018) for training the target model from our evaluation.

---

[3]  Shadow models are used for training the attack models in the **NN**-based attack and selecting the optimal threshold for all metric-based attacks.

[4]  In line with previous work (Choo et al., 2020), we adopt small noise scale (<0.5) for maintaining target model's utility at a decent level, which leads to meaninglessly large $\epsilon$ values.

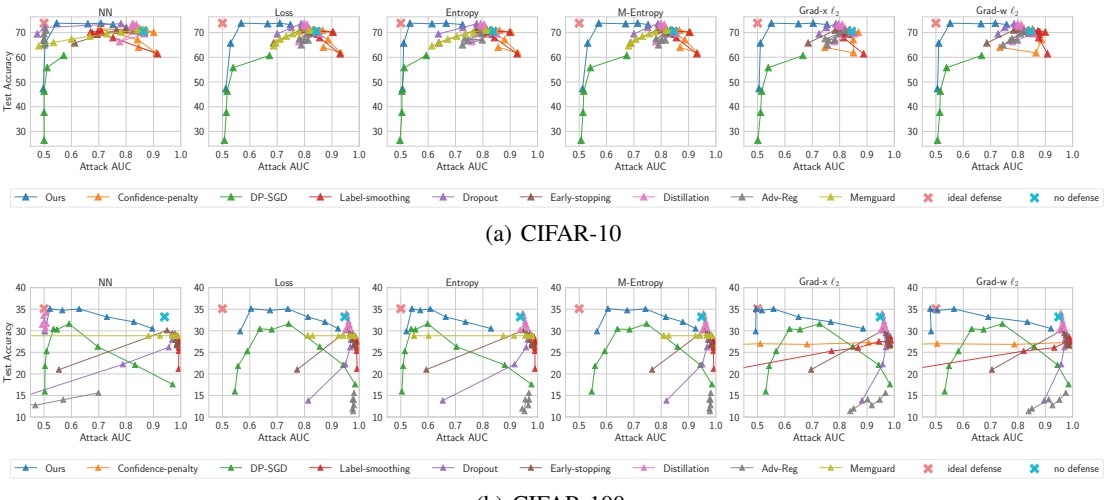

(a) CIFAR-10

(b) CIFAR-100

**Figure 3:** Comparisons of all defense mechanisms on CIFAR-10 and CIFAT-100 dataset with ResNet20 architecture. Each subplot corresponds to one attack method. The x-axis corresponds to the attack AUC *(the lower the better)* while the y-axis is the target model's test accuracy *(the higher the better)*. For visualization purposes, we plot the ideal defense on the *top-left corner*, whose x-coordinate equals 0.5 and y-coordinate is set to be the highest test accuracy among all models.

**Privacy-utility trade-off.** We vary the hyperparameters that best describe the privacy-utility trade-off of each method across their effective ranges (See Appendix B for details) and plot the corresponding privacy-utility curves. We set the attack AUC value (*privacy*) as x-axis and the target model's performance (*utility*) as the y-axis. A better defense exhibits a privacy-utility curve approaching the top-left corner, i.e., high utility and low privacy risk. As shown in Figure 3(a) and 3(b) (and Appendix Figure 9-13), we observe that our method improves the privacy-utility trade-off over baselines for almost all cases: *(i)* Previous state-of-the-art defenses (Memguard and Adv-Reg) are effective for the **NN** attack, but can hardly generalize to the other types of attack, which is also verified in Song & Mittal (2020). Moreover, as a test-time defense, Memguard is not applicable to white-box attacks. In contrast, our method is consistently effective irrespective of the attack model and applicable to all types of attacks. *(ii)* In comparison with other regularization methods, our method showcases significantly better defense effectiveness. Specifically, when compared with Early-stopping, the generally most effective regularization-based defense baseline, our approach decreases the attack AUC (i.e., relative percentage change) by up to 26% on CIFAR-10 and 46% on CIFAR-100 for a same level of utility. *(iii)* DP-SGD is generally the most effective defense baseline, despite the meaningless large $\epsilon$ values, which is consistent with Choo et al. (2020). In comparison with DP-SGD, our method improves the target model's test accuracy by around 16% on CIFAR-10 and 12% on CIFAR-100 (relative percentage increase) across different privacy levels. *(iv)* (See detailed results in Appendix C.9) Our approach is the only one that exhibit consistent defense effectiveness across various data modalities and model architectures, while the best baseline methods can only show effectiveness for at most one data modality (e.g., DP-SGD for images, and Label-smoothing for transaction records).

## 6.3    RELAXLOSS VS. ATTACKS

As can be seen from the privacy-utility curves in Figure 3 and Appendix C.9, our approach is the only one that can consistently defend various MIAs without sacrificing the model utility. We then evaluate *to what extend the attacks can be defended without loss in the model utility*. To this end, we select $\alpha$ corresponds to the model with the *lowest* privacy risk (lowest attack AUC averaged over all attacks), under the constraint that the defended model achieve a top-1 test accuracy *not worse than* the undefended model.

**Utility.** Table 1 summarizes the test accuracy of target models defended with our method. Compared with vanilla training, our method achieves a consistent improvement (up to 7.46%) in terms of utility

(a)  (b)

| Dataset | $N_{train}$ | | CIFAR10 (ResNet20) | | CIFAR10 (VGG11) | | CIFAR100 (ResNet20) | | CIFAR100 (VGG11) | | CH-MNIST | | Texas100 | | Purchase100 | |
|---|---|---|---|---|---|---|---|---|---|---|---|---|---|---|---|---|
| | | | top-1 | top-5 | top-1 | top-5 | top-1 | top-5 | top-1 | top-5 | top-1 | top-5 | top-1 | top-5 | top-1 | top-5 |
| CIFAR-10 | 12000 | wo defense | 70.5 | 96.6 | 73.8 | 97.0 | 33.2 | 63.0 | 41.4 | 67.5 | 77.1 | 99.6 | 52.3 | 82.6 | 89.1 | 99.8 |
| CIFAR-100 | 12000 | with defense | 73.8 | 98.2 | 74.4 | 97.8 | 35.1 | 67.7 | 41.4 | 69.9 | 78.4 | 99.7 | 55.3 | 86.8 | 89.1 | 99.6 |
| CH-MNIST | 1000 | $\Delta$ | 4.68 | 1.66 | 0.81 | 0.82 | 5.72 | 7.46 | 0.00 | 3.56 | 1.69 | 0.10 | 5.74 | 5.08 | 0.00 | -0.20 |
| Texas100 | 13466 | | | | | | | | | | | | | | | |
| Purchase100 | 39465 | | | | | | | | | | | | | | | |

**Table 1:** (a) Size of the target model's training set. (b) Target model's test accuracy (in %) with and without (wo) applying our defense. The *relative* difference ($\Delta$) is in % and the increase is highlighted in green and decrease in red . See Appendix C.3 for more details.

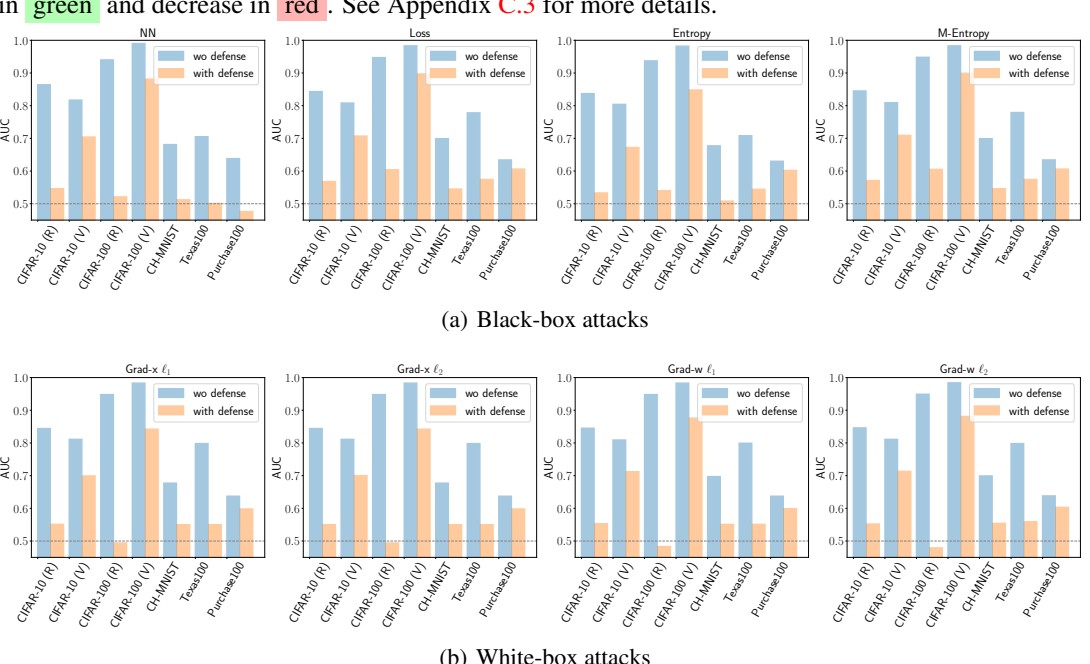

(a) Black-box attacks

(b) White-box attacks

**Figure 4:** Attack AUC on target models trained with and without (wo) applying our defense method. Each subplot is titled with the corresponding attack method's name. (R) and (V) denotes ResNet and VGG network, respectively. The corresponding target model's utility is shown in Table 1.

across different datasets and model architectures, albeit a 0.2% accuracy drop for a saturated top-5 accuracy (99.6% compared to 99.8%).

**Privacy.** Figure 4 shows the membership privacy risk (**AUC**) of the target models in Table 1. We observe that our method is consistently effective for all types of attacks, datasets, and model architectures. In particular, our method consistently reduces the attack AUC: *(i)* to <0.6 for all non-image datasets; *(ii)* from ~0.7 to ~0.55 for CH-MNIST; *(iii)* from >0.8 to ~0.55 for CIFAR-10 (R) and from >0.9 to <0.6 for CIFAR-100 (R). We also include the attack **accuracy** values in Appendix Table 5, which shows our method reduces most attacks to a random-guessing level. We thus conclude that our method improves both target models' utility as well as their resilience against MIAs.

## 6.4 ADAPTIVE ATTACK

We further analyze the robustness of our method against attack's countermeasures. Namely, we consider the situations where attackers have full knowledge about our defense mechanism and the selected hyperparameters, and have tailored the attacks to our defense method. We simulate the adaptive attacks by: *(i)* training shadow models with the same configuration used for training our defended target models in Table 1, *(ii)* simulating the adaptive attacks using the calibrated shadow models. We report the *highest* attack **accuracy** (i.e., worst-case privacy risk) among different *adaptive* attacks in Table 2 (See Appendix C.4 for details). We observe that despite being less effective in defending against adaptive attacks than non-adaptive attacks, our method still greatly decreases the *highest adaptive* attacker's accuracy by 13.6%-37.6% compared to vanilla training.

| | CIFAR10 (ResNet20) | CIFAR10 (VGG11) | CIFAR100 (ResNet20) | CIFAR100 (VGG11) | CH-MNIST | Texas100 | Purchase100 |
|---|---|---|---|---|---|---|---|
| w/o defense | 87.3 | 80.7 | 92.6 | 97.5 | 67.1 | 79.0 | 65.7 |
| w/ defense (non-adaptive) | 50.0 | 50.0 | 50.0 | 50.0 | 50.7 | 50.0 | 50.1 |
| Δ (non-adaptive) | -42.7 | -38.0 | -46.0 | -48.7 | -24.4 | -36.7 | -23.9 |
| w/ defense (adaptive) | 56.0 | 68.2 | 57.8 | 84.2 | 56.6 | 53.8 | 56.0 |
| Δ (adaptive) | -35.9 | -15.5 | -37.6 | -13.6 | -15.6 | -31.9 | -14.8 |

**Table 2:** The *highest* attack **accuracy** (in %) among different adaptive attacks (and the corresponding non-adaptive attack accuracy is shown for reference) evaluated on the target model with (w/) or without (w/o) defense. Δ corresponds to the *relative* difference (in %) in attack accuracy when applying our defense compared to vanilla training. The used target models are the same as in Table 1.

## 6.5 ABLATION STUDY

We study the impact of each component of our approach and plot the results in Figure 5. We observe that while applying posterior flattening alone (without gradient ascent) has limited effects, using it together with gradient ascent indeed improves the model's test accuracy across a wide range of attack AUC, which validates the necessity of all components of our method.

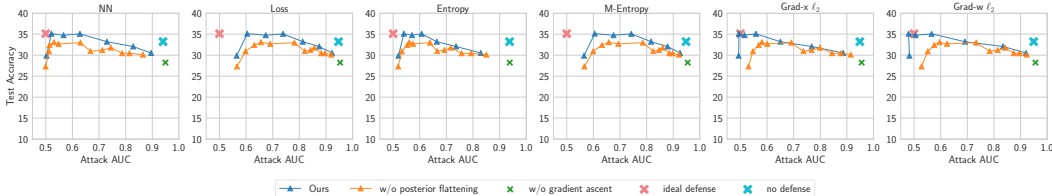

**Figure 5:** Ablation study on CIFAR-100 with ResNet architecture. We validate the necessity of our gradient ascent (Section 4.1) and posterior flattening step (Section 4.2)

## 7 DISCUSSION

**Properties of RelaxLoss.** RelaxLoss enjoys several properties which explain its superiority over existing defense methods. In particular, we provide empirical and analytical evidence showing that in contrast to most existing methods (Appendix C.10), RelaxLoss reduces the generalization gap and spreads out the training loss distributions (Section 5), thereby effectively defeating MIAs. Moreover, we observe that RelaxLoss soften the decision boundaries (Appendix C.11), which contributes to improving model generalization (Zhang et al., 2017; Pereyra et al., 2017).

**Practicality.** We consider the practicality of our method from the following aspects: *(i) Hyperparameter tuning*: Our method involves a single hyperparameter $\alpha$ that controls the trade-off between privacy and utility. A fine-grained grid search on a validation set (i.e., first estimating the privacy-utility trade-off with varying value of $\alpha$, and subsequently selecting the $\alpha$ corresponding to the desired privacy/utility level) allows precise control over the expected privacy/utility level of the target model. *(ii) Computation cost*: Our method incurs negligible additional computation cost when compared with backpropagation in vanilla training (Appendix C.7). In contrast, baseline methods generally suffer from a larger computation burden. For instance, Memguard slows down the inference due to its test-time optimization step, while the training speed of DP-SGD and Adv-Reg is greatly hindered by per-sample gradient computation (Goodfellow, 2015; Dangel et al., 2019) and adversarial update step, respectively.

## 8 CONCLUSION

In this paper, we present *RelaxLoss*, a novel training scheme that is highly effective in protecting against privacy attacks while improving the utility of target models. Our primary insight is that membership privacy risks can be reduced by narrowing the gap between the loss distributions. We validate the effectiveness of our method on a wide range of datasets and models, and evidence its superiority when compared with eight defense baselines which represent previous state of the art. As RelaxLoss exhibits superior protection and performance and is easy to be implemented in various machine learning models, we expect it to be highly practical and widely used.

## ACKNOWLEDGMENTS

This work is partially funded by the Helmholtz Association within the projects "Trustworthy Federated Data Analytics (TFDA)" (ZT-I-OO1 4) and "Protecting Genetic Data with Synthetic Cohorts from Deep Generative Models (PRO-GENE-GEN)" (ZT-I-PF-5-23). Ning Yu was partially supported by Twitch Research Fellowship. We acknowledge Max Planck Institute for Informatics for providing computing resources.

## REPRODUCIBILITY STATEMENT

The authors strive to make this work reproducible. The appendix contains plenty of implementation details. The source code and models are available at GitHub.

## ETHICS STATEMENT

This work does not involve any human subject, nor is our dataset related to any privacy concerns. The authors strictly comply with the ICLR Code of Ethics[5].

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

# Appendix

## Table of Contents

This appendix provides additional support to the main ideas presented in the submission: §A provides additional theoretical analysis giving rise to insights on the foundations of our method. Moreover, as we have conducted a rigorous and broad experimental analysis that goes beyond the key insights presented in the main paper, we provide additional details on the experimental setup in §B and a range of additional evaluation results and discussion in §C.

## A    THEORETICAL ANALYSIS

### A.1    HOW DOES GRADIENT ASCENT STEP INCREASE LOSS VARIANCE?

In this section, we show how the gradient ascent step in RelaxLoss increase the loss variance. We write $\ell$ for the loss instead of $\ell(\boldsymbol{\theta}, \boldsymbol{z}^i)$ for brevity if the dependence is not relevant for our argumentation.

**Theorem A.1.** If $\mathrm{Cov}(\ell, \Delta\ell) > 0$, then the variance of loss distribution $\mathrm{Var}(\ell)$ is increased after a gradient ascent step.

*Proof.* The variance of loss distribution before and after applying gradient ascent step amounts to $\mathrm{Var}(\ell)$ and $\mathrm{Var}(\ell + \Delta\ell)$, respectively. Following from the fact that

$$\mathrm{Var}(\ell + \Delta\ell) = \mathrm{Var}(\ell) + \mathrm{Var}(\Delta\ell) + 2\mathrm{Cov}(\ell, \Delta\ell)$$

and the non-negativity of $\mathrm{Var}(\Delta\ell)$ as well as $\mathrm{Cov}(\ell, \Delta\ell)$, we conclude $\mathrm{Var}(\ell + \Delta\ell) > \mathrm{Var}(\ell)$, i.e., the loss variance will increase.

We focus on the loss increase after the gradient ascent step (i.e., assuming that $\Delta\ell \geq 0$, which holds for most cases, despite the stochasticity) and interpret $\Delta\ell$ as the rate of loss change. The condition $\mathrm{Cov}(\ell, \Delta\ell) > 0$ can be understood as: the larger the loss value is, the faster it changes, and vice versa. This is a reasonable assumption for most training algorithms for achieving convergence. We reason the exact condition and the related assumptions below.

**Condition A.1.** The gradient magnitude (squared $\ell_2$ norm) is positively correlated to the loss value, i.e., $\mathrm{Cov}(\|\nabla\ell\|_2^2, \ell) > 0$. Intuitively, it means the gradient norm tends to decrease as the loss decreases.

We use the cross-entropy loss as an example:

$$\ell_{\mathrm{CE}}(\boldsymbol{\theta}, \boldsymbol{z}_i) = -\sum_{c=1}^{C} y_i^c \log p_i^c \tag{3}$$

The gradient is given by:

$$\nabla\ell_{\mathrm{CE}}(\boldsymbol{\theta}, \boldsymbol{z}_i) = J_{\boldsymbol{\theta} i}(\boldsymbol{p}_i - \boldsymbol{y}_i) \tag{4}$$

where $J_{\boldsymbol{\theta} i}$ represents the jacobian of the logits (before the final softmax layer) w.r.t. the model parameter $\boldsymbol{\theta}$. Once the loss on sample $\boldsymbol{z}_i$ becomes smaller, we have $\|\boldsymbol{p}_i - \boldsymbol{y}_i\|_2 \to 0$, i.e., the prediction get closer to the ground-truth label. By the submultiplicativity of matrix norm and the continuity of the squared function, we then have $\|\nabla\ell_{\mathrm{CE}}(\boldsymbol{\theta}, \boldsymbol{z}_i)\|_2^2 \to 0$, i.e., the gradient norm decreases as the loss value gets smaller. Hence, $\ell_{\mathrm{CE}}(\boldsymbol{\theta}, \boldsymbol{z}_i)$ has the desired property required by Condition A.1.

**Condition A.2.** The change in loss after the gradient ascent step $\Delta\ell$ is linear in the squared gradient norm $\|\nabla\ell\|_2^2$, i.e., $\Delta\ell = c_1\|\nabla\ell\|_2^2 + c_2$ with $c_1, c_2$ the constants quantifying the linear relationship.

**Corollary A.1.** Given the assumption that the gradient of each sample within a batch *(1)* has the same norm and *(2)* has non-negative inner product (i.e., well-aligned) with each other and the gradient alignments remain the same over different batches, the gradient ascent step: $\boldsymbol{\theta}^{(t+1)} = \boldsymbol{\theta}^{(t)} + \tau\nabla\mathcal{L}(\boldsymbol{\theta}^{(t)})$ satisfy the linearity (in Condition A.2) with $c_1 > 0$, where the superscript $t$ corresponds to the $t$-th iteration, and $\nabla\mathcal{L}$ denotes the batch gradient with batchsize $= B$.

*Proof.* This follows from the nature of the first-order gradient-based optimization method. Applying a first-order Taylor-expansion of the sample loss at $\boldsymbol{\theta}^{(t)}$, we obtain:

$$\ell(\boldsymbol{\theta}^{(t+1)}, \boldsymbol{z}_i) = \ell(\boldsymbol{\theta}^{(t)}, \boldsymbol{z}_i) + \tau\langle\nabla\ell(\boldsymbol{\theta}^{(t)}), \nabla\mathcal{L}(\boldsymbol{\theta}^{(t)})\rangle + \mathcal{O}(\tau)$$

$$\Delta\ell = \ell(\boldsymbol{\theta}^{(t+1)}, \boldsymbol{z}_i) - \ell(\boldsymbol{\theta}^{(t)}, \boldsymbol{z}_i) \tag{5}$$

$$= \frac{\tau}{B}\|\nabla\ell(\boldsymbol{\theta}^{(t)}, \boldsymbol{z}_i)\|_2^2 + \frac{\tau}{B}\sum_{j \neq i}\langle\nabla\ell(\boldsymbol{\theta}^{(t)}, \boldsymbol{z}_i), \nabla\ell(\boldsymbol{\theta}^{(t)}, \boldsymbol{z}_j)\rangle + \mathcal{O}(\tau)$$

$$= \frac{\tau}{B}\|\nabla\ell(\boldsymbol{\theta}^{(t)}, \boldsymbol{z}_i)\|_2^2 + \frac{\tau}{B}\sum_{j \neq i}\|\nabla\ell(\boldsymbol{\theta}^{(t)}, \boldsymbol{z}_i)\|_2 \cdot \|\nabla\ell(\boldsymbol{\theta}^{(t)}, \boldsymbol{z}_j)\|_2 \cdot \cos(\alpha_{ij}) + \mathcal{O}(\tau)$$

$$= \frac{\tau}{B}\|\nabla\ell(\boldsymbol{\theta}^{(t)}, \boldsymbol{z}_i)\|_2^2\big(1 + \sum_{j \neq i}\cos(\alpha_{ij})\big) + \mathcal{O}(\tau) \tag{6}$$

where $\cos(\alpha_{ij})$ is the cosine of the angle between gradients of sample $i$ and $j$, and $\mathcal{O}(\tau)$ summarizes the higher-order terms and is regarded as a constant (i.e., $c_2$). Given the assumption that each sample within a batch exhibits well-aligned gradients with the same norm, i.e., $\|\nabla\ell(\boldsymbol{\theta}^{(t)}, \boldsymbol{z}_i)\|_2 = \|\nabla\ell(\boldsymbol{\theta}^{(t)}, \boldsymbol{z}_j)\|_2$ and $\cos(\alpha_{ij}) \geq 0$ for all $j \neq i$, we have Equation 6 and $\frac{\tau}{B}\big(1 + \sum_{j \neq i}\cos(\alpha_{ij})\big) > 0$. Additionally, given that the gradient alignments remain the same over different batches, i.e., $\big(1 + \sum_{j \neq i}\cos(\alpha_{ij})\big)$ is constant for all $i, j$, we have $c_1 = \frac{\tau}{B}\big(1 + \sum_{j \neq i}\cos(\alpha_{ij})\big) > 0$.

**Lemma A.1.** Given the Condition A.1 and A.2, we have the desired property $\mathrm{Cov}(\ell, \Delta\ell)$ by linearity.

*Proof.*

$$\mathrm{Cov}(\ell, \Delta\ell) = \mathrm{Cov}(\ell, c_1\|\nabla\ell\|_2^2 + c_2) \tag{7}$$

$$= c_1\mathrm{Cov}(\ell, \|\nabla\ell\|_2^2) > 0 \tag{8}$$

where Equation 7 and 8 are yielded by using Condition A.2 and A.1, respectively.

## A.2 HOW DOES RELAXLOSS AFFECT MIA?

In this section, we show how RelaxLoss affects the optimal MIA $\mathcal{A}_{opt}$ (measured by its AUC value). We exploit the following results for relating the attack AUC to a statistical distance between the loss distributions.

We first regard the MIA as a binary hypothesis testing problem with the null $H_0$ and alternate hypothesis $H_1$ defined as follows:

$$H_0 : \quad z_i \text{ is a member sample, i.e., } z_i \in \mathcal{D}_{\text{train}}$$
$$H_1 : \quad z_i \text{ is a non-member sample, i.e., } z_i \notin \mathcal{D}_{\text{train}}$$

The attacker need to make a decision on whether the query sample came from $\mathcal{D}_{\text{train}}$ based on a rejection region $S_{\text{reject}}$. As discussed in Section 4.1, under a posterior assumption on the model parameter, $S_{\text{reject}}$ for the optimal attack $\mathcal{A}_{opt}$ (Sablayrolles et al., 2019) fully depends on the sample loss, i.e., $\mathcal{A}_{opt}$ rejects the null hypothesis if $\ell(\theta, z_i) \in S_{\text{reject}}$. The type I error (i.e., the $H_0$ is true but rejected) is defined as $\mathcal{P}(\ell(\theta, \mathcal{D}_{\text{train}}) \in S_{\text{reject}})$, and the type II error (i.e., the $H_0$ is false but retained) is defined as $\mathcal{P}(\ell(\theta, \overline{\mathcal{D}}_{\text{train}}) \in \overline{S}_{\text{reject}})$.

**Theorem A.2.** (Kairouz et al., 2015; Lin et al., 2018; 2021) Let TP and FP denote the true positive rate ($1-$ type II error) and false positive rate (type I error) of $\mathcal{A}_{opt}$ respectively, their relation to the total variation distance between the loss distributions $D_{\text{TV}}(P, Q)$ is quantified as follows (See Kairouz et al. (2015) Appendix A for the derivation):

$$\text{TP} \leq \text{FP} + \min\{D_{\text{TV}}(P, Q), 1 - \text{FP}\} \tag{9}$$

where $P$ and $Q$ denote the distribution of the training loss $\ell(\theta, \mathcal{D}_{\text{train}})$ and the testing loss $\ell(\theta, \overline{\mathcal{D}}_{\text{train}})$, respectively.

The ROC curve is obtained by plotting the largest achievable true positive (TP) rate on the vertical axis against the false positive (FP) rate on the horizontal axis, while the AUC value corresponds to a summation over all pairs of TP and FP.

**Corollary A.2.** The AUC value can be upper bounded as follows (Lin et al. (2021) Corollary 1):

$$\text{AUC} \leq -\frac{1}{2}D_{\text{TV}}(P, Q)^2 + D_{\text{TV}}(P, Q) + 1/2 \tag{10}$$

For ease of analysis, we then upper bound the total variation distance via the Hellinger distance, which is symmetric and has a closed-form expression for common distributions such as Gaussian.

**Theorem A.3.** (Steerneman, 1983)The total variance distance can be upper bounded by the Hellinger distance:

$$D_{\text{TV}}(P, Q) \leq \sqrt{2}D_{\text{H}}(P, Q) \tag{11}$$

where the Hellinger distance satisfies $0 \leq D_{\text{H}}(P, Q) \leq 1$

For Gaussian distributions, the Hellinger distance has a closed form:

$$D_{\text{H}}^2(P, Q) = 1 - \sqrt{\frac{2\sigma_1\sigma_2}{\sigma_1^2 + \sigma_2^2}} \exp\left(-\frac{1}{4}\frac{(\mu_1 - \mu_2)^2}{\sigma_1^2 + \sigma_2^2}\right) \tag{12}$$

where $\mu_1, \mu_2$ denote the mean and $\sigma_1, \sigma_2$ denote the variance of $P$ and $Q$.

Let $c = \sigma_2/\sigma_1$ denote the ratio of the training and testing loss variance, we see that $D_{\text{H}}(P, Q)$ is fully characterized by: *(i)* the value of the training loss variance $\sigma_1^2$, *(ii)* the squared distance between the mean $(\mu_1 - \mu_2)^2$, and *(iii)* the variance ratio $c$:

$$D_{\text{H}}^2(P, Q) = 1 - \underbrace{\sqrt{\frac{2c}{1 + c^2}}}_{(*)} \underbrace{\exp\left(-\frac{1}{4}\frac{(\mu_1 - \mu_2)^2}{(1 + c^2)\sigma_1^2}\right)}_{(**)} \tag{13}$$

Our approach decreases the $(\mu_1 - \mu_2)^2$ and increases $\sigma_1^2$ (Section 5) as well, both of which lead to a decrease of the Hellinger distance, and thus decreases the upper bound of the attacker AUC as desired.

It remains to consider how our approach will change $c$ and how the change in $c$ will affect the Hellinger distance. First, we observe that $c \geq 1$, i.e., the testing distribution has larger variance than the training distribution (See Appendix C.10). Moreover, $c$ gets closer to 1 when applying our approach (See Figure 1 in the main paper). As a result, $(*)$ will increase (Corollary A.3) and $(**)$ will decrease (Corollary A.4).

**Corollary A.3.** If $c' \geq c \geq 1$, then $\sqrt{\frac{2c}{1+c^2}} \geq \sqrt{\frac{2c'}{1+c'^2}}$

*Proof.* Let $f(c) = \sqrt{\frac{2c}{1+c^2}}$. We have $f'(c) = \frac{1-c^2}{\sqrt{2c}(c^2+1)^{3/2}}$. It is obvious that $f$ has critical point at $c = 1$, i.e., $f'(1) = 0$ and $f'(c) \leq 0$.

**Corollary A.4.** For fixed value of $\mu_1, \mu_2$ and $\sigma_1$, if $c' \geq c \geq 1$, then

$$\exp\left(-\frac{1}{4}\frac{(\mu_1 - \mu_2)^2}{(1+c^2)\sigma_1^2}\right) \leq \exp\left(-\frac{1}{4}\frac{(\mu_1 - \mu_2)^2}{(1+c'^2)\sigma_1^2}\right)$$

*Proof.* It is obvious as $c$ occurs in the denominator inside the exponential term.

Additionally, we notice that the change in the $(*)$ dominates in most cases: the $(**)$ term commonly has value within $[0.9, 1.0]$, while the $(*)$ term changes from $10^{-3}$ to 1 when our approach is applied. Therefore, under a Gaussian assumption of the loss distributions, our method can decrease the Hellinger distance between the distributions, thereby reducing an upper bound of the attack AUC.

# B EXPERIMENT SETUP

## B.1 DATASETS

**CIFAR-10 (Krizhevsky et al., 2009)** is a dataset of 60k color images with shape $32 \times 32 \times 3$. Each image corresponds to a label of 10 classes which categorizes the object inside the image. Following the standard preprocessing procedure [6], we normalize the image pixel value to have zero mean and unit standard deviation.

**CIFAR-100 (Krizhevsky et al., 2009)** consists of 60k color images of size $32 \times 32 \times 3$ in 100 classes. Same as for the CIFAR-10 dataset, we perform mean-subtraction and standardization.

**CH-MNIST (Kather et al., 2016)** contains 5000 greyscale images of 8 different types of tissues from patients with colorectal cancer. We obtain the preprocessed dataset from Kaggle [7] and use images of size 28×28 for our experiments. All images are normalized to $[-1, 1]$.

**Texas100** contains medical records of 67,330 patients published by the Texas Department of State Health Services [8]. Each patient's record contains 6,169 binary features (such as diagnosis, generic information, and procedures the patient underwent) and is labeled by its most suitable procedure (among the 100 most frequent ones). We use the preprocessed data provided by Shokri et al. (2017); Song & Mittal (2020)[9].

**Purchase100** is a dataset of customers' shopping records released by the Kaggle Acquire Valued Shoppers Challenge [10]. We use the preprocessed version provided by Shokri et al. (2017); Song & Mittal (2020)[9], which contains 197,324 data samples. Each sample, representing one user's purchase history, consists of 600 binary features. Each feature denotes the presence of one product in the corresponding user's purchase history. The data is clustered into 100 classes of different purchase styles. The classification task is to predict the purchase style given the 600 binary features.

We summarize all datasets in details in Table 3.

---

[6]   https://pytorch.org/hub/pytorch_vision_resnet/
[7]   https://www.kaggle.com/kmader/colorectal-histology-mnist
[8]   https://www.dshs.texas.gov/THCIC/Hospitals/Download.shtm
[9]   https://github.com/inspire-group/membership-inference-evaluation
[10]  https://www.kaggle.com/c/acquire-valued-shoppers-challenge

| Dataset | Data type | Feature type | Feature dimension | $N_{\text{total}}$ | $N_{\text{train}}/N_{\text{test}}$ (target model) | $N_{\text{train}}/N_{\text{test}}$ (shadow model) |
|---|---|---|---|---|---|---|
| CIFAR-10 | color image | numerical | 3072 | 60000 | 12000 | 12000 |
| CIFAR-100 | color image | numerical | 3072 | 60000 | 12000 | 12000 |
| CH-MNIST | grayscale image | numerical | 784 | 5000 | 1000 | 1000 |
| Texas100 | medical record | categorical | 6169 | 67330 | 13466 | 13466 |
| Purchase100 | purchase record | categorical | 600 | 197324 | 39465 | 39464 |

**Table 3:** Summary of datasets. $N_{\text{total}}$ denotes the total dataset size. $N_{\text{train}}$ and $N_{\text{test}}$ are the size of the training and testing set, respectively.

## B.2 MODEL ARCHITECTURES

For CIFAR-10 and CIFAR-100 datasets, we use a 20-layer ResNet and an 11-layer VGG architecture [11]. For CH-MNIST, we adopt a 20-layer ResNet. And for the non-image datasets, we adopt the same architecture as used in Nasr et al. (2018)[12]: a 4-layer fully-connected neural network with layer size [1024, 512, 256, 100] for Purchase100, and a 5-layer fully-connected neural network with layer size [2048, 1024, 512, 256, 100] for Texas100.

## B.3 IMPLEMENTATION DETAILS

We apply SGD optimizer with momentum=0.9 and weight-decay=1e-4 by default. We set the initial learning rate $\tau = 0.1$ and drop the learning rate by a factor of 10 at each decay epoch [11]. We list below the decay epochs in square brackets and the total number of training epochs are marked in parentheses: CIFAR-10 and CIFAR-100 [150,225] (300); CH-MNIST [40,60] (80); Texas100 and Purchase100 [50,100] (120). Additionally, we adopt the following techniques for improved performance across heterogeneous data modalities: we restrict the scope of posterior flattening to *incorrect predictions* for natural image datasets (CIFAR-10 and CIFAR-100); and we further suppress the target posterior scores of *ground-truth* class $p^{gt}$ to small values ($\leq 0.3$) during the posterior flattening step on non-image data (Texas100 and Purchase100). By default, no data augmentation is applied.

## B.4 REQUIRED RESOURCES

All our models and methods are implemented in PyTorch. Our experiments are conducted with Nvidia Tesla V100 and Quadro RTX8000 GPUs. Our method introduces minimal changes and negligible additional cost compared with vanilla training and thus can be flexibly integrated into any deep learning framework without imposing specific constraints on the required hardware resources.

## B.5 DEFENSE METHODS

**Early-stopping.** The basic idea behind Early-stopping is to truncate training before a model starts to overfit. In our experiments, we save target models' checkpoints at varying numbers of training epochs and subsequently evaluate the attack AUC and test accuracy of each model checkpoint. We set checkpoints at the following epochs: $[25, 50, 75, 100, 125, 150, 175, 200, 225, 250, 275]$ for CIFAR-10 and CIFAR-100 datasets; $[5, 10, 15, 20, 25, 30, 40, 50]$ for CH-MNIST dataset; $[10, 20, 30, 40, 50, 60, 70, 80, 90, 100, 110]$ for Texas100 and Purchase100 datasets.

**Dropout.** Dropout prevents co-adaptation of feature detectors by randomly masking out a set of neurons in the networks, thereby alleviating model overfitting. In our experiments, we apply dropout to the last fully-connected layer of each target model and evaluate across a wide range of dropout rates (over $[0.1, 0.3, 0.5, 0.7, 0.9]$).

**Label-smoothing.** Label-smoothing prevents overconfident predictions by incorporating a regularization term into the training objective that penalizes the distance (measured by the KL-divergence) between the model predictions and the uniform distribution. The objective is formularized as follows

$$\mathcal{L} = \alpha \cdot D_{\text{KL}}(\mathcal{U} \, \| \, p_{\boldsymbol{\theta}}(\boldsymbol{y}|\boldsymbol{x})) + (1 - \alpha) \cdot \mathcal{L}_{\text{CE}}(\boldsymbol{\theta}, \boldsymbol{z}) \tag{14}$$

---

[11] https://github.com/bearpaw/pytorch-classification
[12] https://github.com/SPIN-UMass/ML-Privacy-Regulization

where $D_{\mathrm{KL}}$ is the KL-divergence; $\mathcal{U}$ denotes the uniform distribution; $p_{\boldsymbol{\theta}}(\boldsymbol{y}|\boldsymbol{x})$ denotes the output prediction. $\alpha$ is a hyper-parameter with range $[0, 1]$ that balances the cross-entropy loss $\mathcal{L}_{\mathrm{CE}}$ and the regularization term. We vary the $\alpha$ across its full range for plotting the privacy-utility curves.

**Confidence-penalty.** Confidence-penalty regularizes models by penalizing low entropy output distributions. This is achieved via an entropy regularization term in the objective:

$$\mathcal{L} = -\alpha \cdot H(p_{\boldsymbol{\theta}}(\boldsymbol{y}|\boldsymbol{x})) + \mathcal{L}_{\mathrm{CE}}(\boldsymbol{\theta}, \boldsymbol{z}) \tag{15}$$

$$\text{where} \quad H(p_{\boldsymbol{\theta}}(\boldsymbol{y}|\boldsymbol{x})) = -\sum_{c=1}^{C} p_{\boldsymbol{\theta}}(y^c|\boldsymbol{x}) \log(p_{\boldsymbol{\theta}}(y^c|\boldsymbol{x})) \tag{16}$$

$H$ represents the entropy of the output prediction, and $\alpha$ is a hyper-parameter that controls the importance of the entropy regularization term. Consistent with the original paper (Pereyra et al., 2017), we vary the $\alpha$ over $[0.1, 0.3, 0.5, 1.0, 2.0, 4.0, 8.0]$.

**Distillation.** Knowledge distillation stands for a general process of transferring knowledge from a set of teacher model(s) to a student model. To focus our investigation on the effect of the distillation operation itself, we use self-distillation (Zhang et al., 2019) in our experiments, i.e., we train the student model to match a single teacher model with the same architecture. The objective for training the student model (i.e., the target model) is:

$$\mathcal{L} = \alpha T^2 \cdot D_{\mathrm{KL}}(\widetilde{p}_{\boldsymbol{\theta}_s}(\boldsymbol{y}|\boldsymbol{x}) \,\|\, \widetilde{p}_{\boldsymbol{\theta}_t}(\boldsymbol{y}|\boldsymbol{x})) + (1 - \alpha) \cdot \mathcal{L}_{\mathrm{CE}}(\boldsymbol{\theta}, \boldsymbol{z}) \tag{17}$$

$$\text{where} \quad \widetilde{p}_{\boldsymbol{\theta}}(\boldsymbol{y}|\boldsymbol{x})^c = \frac{\exp(f(\boldsymbol{\theta}, \boldsymbol{x})^c / T)}{\sum_{c'} \exp(f(\boldsymbol{\theta}, \boldsymbol{x})^{c'} / T)} \tag{18}$$

The KL-divergence term $D_{\mathrm{KL}}$ targets at minimizing discrepancy between the softened student $\widetilde{p}_{\boldsymbol{\theta}_s}(\boldsymbol{y}|\boldsymbol{x})$ and teacher prediction $\widetilde{p}_{\boldsymbol{\theta}_t}(\boldsymbol{y}|\boldsymbol{x})$. $T$ denotes the temperature scaling factor that controls the degree of softening. $\alpha$ is a hyper-parameter that balances the KL-divergence and the normal cross-entropy $\mathcal{L}_{\mathrm{CE}}$ term. To determine the hyper-parameter that best describes the privacy-utility trade-off, we conduct preliminary experiments and investigate the effect of $\alpha$ and $T$ independently. By fixing one and changing the other, we observe similar results in terms of the privacy-utility trade-off. Following practical standards, we then fix the $\alpha$=0.5 and vary the temperature $T$ over $[1, 2, 5, 10, 20, 50, 100]$ for plotting the privacy-utility curves.

**DP-SGD.** DP-SGD enforces privacy guarantees by modifying the optimization process. It consists of two steps: *(i)* clipping the gradients to have a $L_2$-norm upper-bounded by $C$ at each training step; *(ii)* injecting random noise to the gradients before performing update steps. We adopt the implementation provided by the Opacus library [13]. We tune the clipping bound $C$ when fixing the noise scale to 0.1, as suggested by the official documents. To plot the privacy-utility curves, we vary the noise scale with a fixed pre-selected clipping bound.

**Memguard.** Memguard modifies the output predictions of pre-trained target models during test-time, i.e., output predictions are perturbed by adversarial noise to fool a surrogate attack model. Following the official implementation[14], we adopt the same architecture for the surrogate and the real **NN** attack model, and use the complete logits prediction as input to the attack models. Each surrogate attack model is trained on the target model's predictions when inputting the target model's training data (used as member samples) and a separate hold-out set data (used as non-member samples). The privacy-utility trade-off is fully determined by the magnitude of the adversarial noise (measured by $L_1$-norm). We plot the privacy-utility curves by increasing the noise magnitude from 0 until the **NN** attack has been defended (i.e., attack AUC $\approx 0.5$).

**Adv-Reg.** Adv-Reg incorporates an adversarial objective when training the target model: the target model is trained to minimize a weighted sum of the cross-entropy loss and the adversarial loss (obtained from a surrogate attack). Same as in Memguard, each surrogate attack model is trained on the target model's training data and a separate hold-out set data. We follow the official implementation[12] and vary the weight $\alpha$ (=1.0 by default) of the adversarial loss across $[0.8, 1.0, 1.2, 1.4, 1.6, 1.8]$ for plotting the privacy-utility curves.

---

[13]  https://github.com/pytorch/opacus
[14]  https://github.com/jjy1994/MemGuard

## C  ADDITIONAL RESULTS AND DISCUSSION

### C.1  LIMITATIONS AND FUTURE WORKS

Although RelaxLoss is empirically proven effective for improving target models' utility, it is generally hard to explain such improvement, as understanding the generalization ability is still an open problem. As an attempt, we conduct experiments on toy datasets and attribute the improvement to a flat decision boundary (See Appendix C.11). A thorough investigation into how each individual components of our approach affects generalization are left for our future works. In addition, the assumptions of model parameters (Sablayrolles et al., 2019) which are made for the optimal attack demand further validation.

### C.2  CORRELATION BETWEEN LOSS VARIANCE AND ATTACK AUC

We plot the attack AUC values versus the training loss variance in Figure 6. Each point in the figure corresponds to one target model trained with different defense mechanisms. We indeed observe a negative correlation between the loss variance and the AUC value of both the black-box and white-box attacks, which supports Section 5 in the main paper.

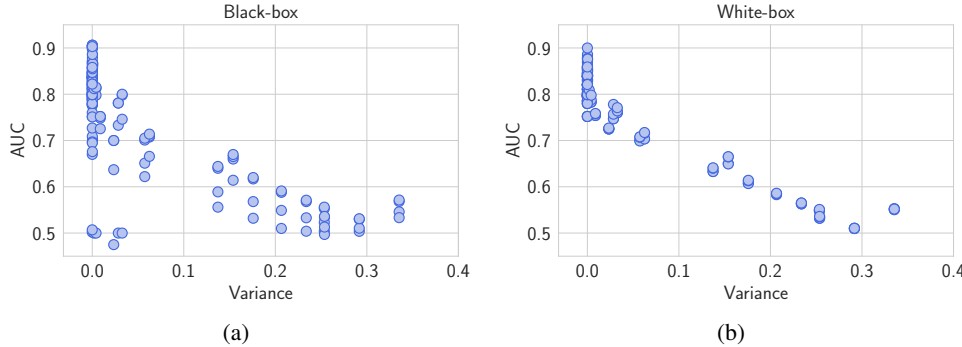

**Figure 6:** Correlation between the training set loss variance and the MIA performance on CIFAR-10 (ResNet20). The Pearson's correlation coefficients equal to: (a) -0.77 for black-box attacks; (b) -0.94 for white-box attacks; and -0.85 if considering black-box and white-box attacks together.

### C.3  RELAXLOSS VS. ATTACKS

Supplementary to Table 1 in the main paper, Table 4 summarizes the top-1 *training* and *test* accuracy as well as the *generalization gap* (i.e., difference between the top-1 training and test accuracy) of target models with or without being defended via our method. We observe that our approach, as desired, reduces the generalization gap and is still able to achieve high performance.

| | CIFAR-10 (ResNet20) | | | CIFAR-10 (VGG11) | | | CIFAR-100 (ResNet20) | | | CIFAR-100 (VGG11) | | | CH-MNIST | | | Texas100 | | | Purchase100 | | |
|---|---|---|---|---|---|---|---|---|---|---|---|---|---|---|---|---|---|---|---|---|---|
| | train | test | gap | train | test | gap | train | test | gap | train | test | gap | train | test | gap | train | test | gap | train | test | gap |
| wo defense | 100 | 70.5 | 29.5 | 100 | 73.8 | 26.2 | 100 | 33.2 | 66.8 | 100 | 41.4 | 58.6 | 99.0 | 77.1 | 21.9 | 99.9 | 52.3 | 47.6 | 100 | 89.1 | 10.9 |
| with defense | 87.0 | 73.8 | 13.2 | 99.5 | 74.4 | 25.1 | 52.7 | 35.1 | 17.6 | 99.7 | 41.4 | 58.3 | 90.1 | 78.4 | 11.7 | 67.1 | 55.3 | 11.8 | 99.8 | 89.1 | 10.7 |
| Δ | -13.0 | 3.3 | -16.3 | -0.5 | 0.6 | -1.1 | -47.3 | 1.9 | -49.2 | -0.3 | 0.0 | -0.3 | -8.9 | 1.3 | -10.2 | -32.8 | 3.0 | -35.8 | -0.2 | 0.0 | -0.2 |
| selected $\alpha$ | 1 | | | 0.4 | | | 3 | | | 0.5 | | | 0.2 | | | 2.5 | | | 0.8 | | |

**Table 4:** The top-1 *training* and *test* accuracy as well as the generalization *gap* (in %) of the target models with or without (wo) applying our defense. Δ corresponds to the *absolute* difference after applying our defend method (in %). We also include the selected value of $\alpha$. This is supplementary to Table 1 in the main paper.

In Table 5, we show the attack **accuracy** values on target models trained with and without (wo) applying our defense method, which is supplementary to Table 1 and Figure 4 in the main paper. Same as in previous work (Song & Mittal, 2020)[9], the attack's decision threshold is selected to be the one that yields the best attack accuracy on undefended shadow models. We observe that our approach effectively reduces the attack accuracy to a random-guessing level (around 50%) for most cases. In

| | | Entropy | M-Entropy | Loss | NN | Grad-x $\ell_1$ | Grad-x $\ell_2$ | Grad-w $\ell_1$ | Grad-w $\ell_2$ |
|---|---|---|---|---|---|---|---|---|---|
| CIFAR-10 (ResNet20) | wo defense | 86.5 | 87.3 | 86.9 | 82.5 | 87.5 | 87.5 | 87.8 | 87.8 |
| | with defense | 50.0 | 50.0 | 50.0 | 49.9 | 50.0 | 50.0 | 49.9 | 50.0 |
| | Δ | -42.2 | -42.7 | -42.5 | -39.5 | -42.9 | -42.9 | -43.2 | -43.1 |
| CIFAR-10 (VGG11) | wo defense | 80.1 | 80.7 | 80.6 | 76.1 | 81.5 | 81.3 | 82.7 | 82.9 |
| | with defense | 50.0 | 50.0 | 50.0 | 50.0 | 50.0 | 50.0 | 50.0 | 50.0 |
| | Δ | -37.6 | -38.0 | -38.0 | -34.3 | -38.7 | -38.5 | -39.5 | -39.7 |
| CIFAR-100 (ResNet20) | wo defense | 91.8 | 92.1 | 92.6 | 87.0 | 93.7 | 93.7 | 94.6 | 94.7 |
| | with defense | 50.0 | 50.0 | 50.0 | 50.0 | 50.0 | 50.0 | 50.0 | 50.0 |
| | Δ | -45.5 | -45.7 | -46.0 | -42.5 | -46.6 | -46.6 | -47.1 | -47.2 |
| CIFAR-100 (VGG11) | wo defense | 97.1 | 97.5 | 97.4 | 98.1 | 98.5 | 98.4 | 98.9 | 98.9 |
| | with defense | 50.0 | 50.0 | 50.0 | 50.6 | 50.1 | 50.1 | 50.6 | 50.4 |
| | Δ | -48.5 | -48.7 | -48.7 | -48.4 | -49.1 | -49.1 | -48.8 | -49.0 |
| CH-MNIST | wo defense | 55.5 | 56.7 | 56.7 | 63.6 | 67.6 | 67.7 | 67.1 | 66.4 |
| | with defense | 49.9 | 52.3 | 50.9 | 50.3 | 52.2 | 51.7 | 50.7 | 49.9 |
| | Δ | -10.1 | -7.8 | -10.2 | -20.9 | -22.8 | -23.6 | -24.4 | -24.8 |
| Texas100 | wo defense | 70.3 | 79.0 | 79.0 | 63.8 | 78.5 | 78.5 | 78.4 | 78.3 |
| | with defense | 50.0 | 50.0 | 50.0 | 50.0 | 52.0 | 53.0 | 51.1 | 50.0 |
| | Δ | -28.9 | -36.7 | -36.7 | -21.6 | -33.8 | -32.5 | -34.8 | -36.1 |
| Purchase100 | wo defense | 63.9 | 64.8 | 64.7 | 62.6 | 65.8 | 65.7 | 65.8 | 65.7 |
| | with defense | 50.0 | 50.0 | 50.0 | 49.6 | 52.3 | 52.2 | 50.1 | 50.1 |
| | Δ | -21.8 | -22.8 | -22.7 | -20.8 | -20.5 | -20.5 | -23.9 | -23.9 |

**Table 5:** The attacker **accuracy** (in %) evaluated on the target models with and without (wo) applying our defense. Δ corresponds to the *relative* difference after applying our defend method (in %). All the thresholds are selected with undefended shadow models trained with shadow dataset.

particular, we find that the selected decision thresholds is highly biased in certain cases s.t. all the queried samples are predicted to be positive (or negative), which leads to exactly 50% accuracy.

## C.4 ADAPTIVE ATTACK

In Table 6, we show the **accuracy** of adaptive (a.) attacks on target models trained with (w/) applying our defense method, which is supplementary to Section 6.4 in the main paper. For thresholding-based attacks, the attack's decision threshold is selected to be the one that yields the best attack accuracy on the shadow models (which is trained with exactly the same configuration as our defended target models in Table 1). And for the NN-based attack, we use the complete logits prediction from the pre-trained shadow models as features to train the adaptive attack models (modeled as a NN). We observe that our approach consistently reduces the attack accuracy for all cases, though the reduction is less significant compared to non-adaptive attacks shown in Table 5.

| | | Entropy | M-Entropy | Loss | NN | Grad-x $\ell_1$ | Grad-x $\ell_2$ | Grad-w $\ell_1$ | Grad-w $\ell_2$ |
|---|---|---|---|---|---|---|---|---|---|
| CIFAR10 (ResNet20) | w/ defense (a.) | 52.5 | 56.0 | 56.0 | 54.2 | 53.5 | 53.3 | 54.0 | 53.8 |
| | Δ | -39.3 | -35.9 | -35.6 | -34.3 | -38.9 | -39.1 | -38.5 | -38.7 |
| CIFAR10 (VGG11) | w/ defense (a.) | 64.4 | 68.2 | 67.8 | 66.6 | 66.2 | 66.4 | 67.4 | 68.0 |
| | Δ | -19.6 | -15.5 | -15.9 | -12.5 | -18.8 | -18.3 | -18.5 | -18.0 |
| CIFAR100 (ResNet20) | w/ defense (a.) | 52.2 | 57.8 | 57.8 | 53.6 | 50.1 | 50.1 | 50.0 | 50.1 |
| | Δ | -43.1 | -37.2 | -37.6 | -38.4 | -46.5 | -46.5 | -47.1 | -47.1 |
| CIFAR100 (VGG11) | w/ defense (a.) | 78.9 | 84.2 | 84.0 | 83.4 | 78.2 | 78.2 | 81.6 | 82.2 |
| | Δ | -18.7 | -13.6 | -13.8 | -15.0 | -20.6 | -20.5 | -17.5 | -16.9 |
| CH-MNIST | w/ defense (a.) | 50.7 | 53.9 | 53.4 | 50.9 | 55.8 | 55.7 | 56.6 | 56.4 |
| | Δ | -8.6 | -4.9 | -5.8 | -20.0 | -17.5 | -17.7 | -15.6 | -15.1 |
| Texas100 | w/ defense (a.) | 51.6 | 53.8 | 53.8 | 52.1 | 51.9 | 51.9 | 51.8 | 53.4 |
| | Δ | -26.6 | -31.9 | -31.9 | -18.3 | -33.9 | -33.9 | -33.9 | -31.8 |
| Purchase100 | w/ defense (a.) | 54.0 | 54.8 | 54.8 | 54.5 | 55.4 | 55.4 | 55.6 | 56.0 |
| | Δ | -15.5 | -15.4 | -15.3 | -12.9 | -15.8 | -15.7 | -15.5 | -14.8 |

**Table 6:** The **accuracy** (in %) of adaptive (a.) attacks evaluated on the target models with (w/) applying our defense. Δ corresponds to the *relative* difference (in %) attack accuracy when applying our defend method compared to vanilla training. The selected target models are the same as in Table 1.

## C.5 GENERALIZATION GAP

We show the training and testing accuracy (and the generalization gap) when applying RelaxLoss with varying value of $\alpha$ in Figure 7. We observe that increasing the value of $\alpha$ will reduce the generalization gap. Moreover, RelaxLoss with a reasonable value of $\alpha$ can even improves the test accuracy of vanilla training.

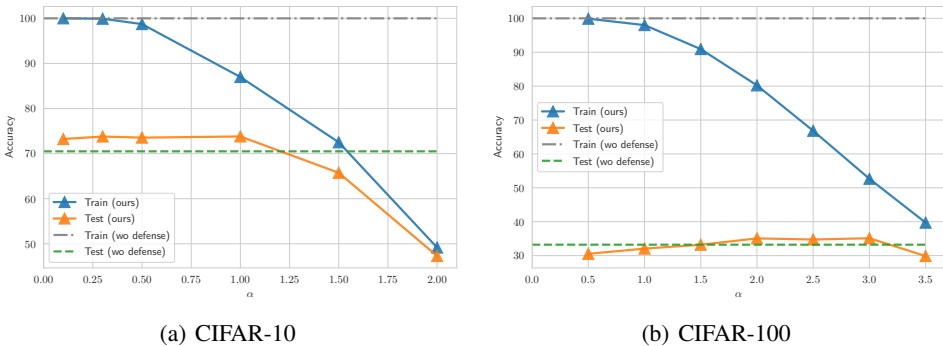

(a) CIFAR-10                                    (b) CIFAR-100

**Figure 7:** Training and testing accuracy (y-axis) with varying value of $\alpha$ (x-axis) on CIFAR-10 (ResNet20) and CIFAR-100 (ResNet20) datasets. We plot the training and testing accuracy of vanilla training (wo defense) in dashed lines for reference.

### C.6 COMPATIBILITY WITH DATA AUGMENTATION

Additionally, we investigate the effectiveness of our approach when data augmentation is applied. Following practical standard, we apply *random cropping* and *random flipping* when training the target models on CIFAR-100 dataset. As illustrated in Figure 8, RelaxLoss is compatible with standard data augmentation techniques: our approach enjoys the performance boost introduced by data augmentation while retaining its effectiveness in defending MIAs.

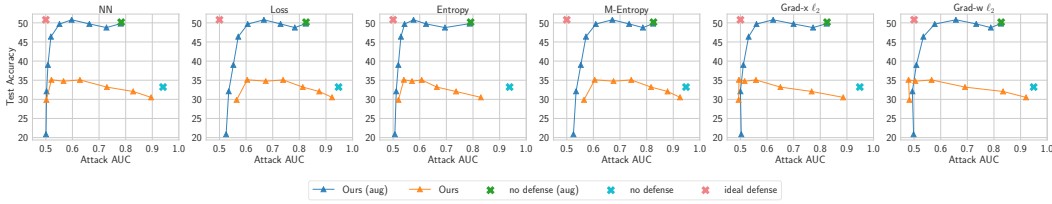

**Figure 8:** Effect of data augmentation (denoted as "aug") on CIFAR-100 (ResNet20). When jointly applied with data augmentation, our approach shows consistent effectiveness in improving the MIA resistance and model utility.

### C.7 COMPUTATIONAL COMPLEXITY

The additional computation cost of RelaxLoss scales as $\mathcal{O}(BC)$ ($B$: batch size; $C$: number of classes), which includes: *(i)* softlabel construction of cost $\mathcal{O}(BC)$; and *(ii)* computation of the cross-entropy loss on the softlabel $\mathcal{O}(BC)$. Note that we reuse the prediction $\mathbf{p}_i$ generated by the previous forward-pass and thus no additional forward (nor backward) pass is required. Compared to the forward and backward pass, which is of magnitude at least $\mathcal{O}(BNL)$ ($N$: number of neurons per layer; $L$: number of layers), the additional costs of RelaxLoss are negligible as $NL$ (roughly the total amount of neurons of the whole network) is much larger than $C$ (the number of neurons of the last layer).

### C.8 EFFECT ON DIFFERENT CLASSES OF INDIVIDUALS

To analyze the effect of RelaxLoss on different individuals, we conduct additional experiments on Texas and Purchase datasets which consist of 100 classes with non-uniform class distribution (i.e., the proportion of each class ranges from 0.35% to 4.5% for Texas, and 0.05%-2.6% for Purchase) and evaluate the attack performance on each class separately.

In Table 7, we show the 10 *highest* **Attack AUC** (in increasing order) among all the classes, which can be regarded as the estimated *worst-case* privacy risk of different classes of individuals. As can be seen from the tables, applying our defense method effectively reduces the Top-10 Attack AUC (i.e., worst-case privacy risk), and the effectiveness is *consistent* on each dataset across different attack

methods, with which we conclude that our method, despite the nonuniformity, does defend MIAs for different individuals.

(a) Texas

| Atttack methods | with/wo defense | Top-10 Attack AUC |
|---|---|---|
| Loss | wo defense | 0.985, 0.987, 0.988, 0.989, 0.994, 0.994, 0.995, 0.996, 0.998, 0.999 |
| | with defense | 0.717 , 0.719, 0.721, 0.722, 0.724, 0.739, 0.741, 0.743, 0.752, 0.761 |
| Entropy | wo defense | 0.846, 0.847, 0.851, 0.851, 0.854, 0.864, 0.865, 0.868, 0.879, 0.880 |
| | with defense | 0.617, 0.619, 0.625, 0.625, 0.636, 0.636, 0.648, 0.679, 0.733, 0.747 |
| M-Entropy | wo defense | 0.985, 0.987, 0.989, 0.990, 0.992, 0.993, 0.994, 0.996, 0.997, 0.999 |
| | with defense | 0.717, 0.718, 0.722, 0.722, 0.724, 0.741, 0.742, 0.742, 0.754, 0.761 |
| Grad-x l2 | wo defense | 0.837, 0.837, 0.844, 0.848, 0.850, 0.852, 0.859, 0.865, 0.903, 0.943 |
| | with defense | 0.644, 0.650, 0.653, 0.655, 0.657, 0.666, 0.679, 0.691, 0.723, 0.744 |
| Grad-w l2 | wo defense | 0.850, 0.852, 0.853, 0.856, 0.862, 0.862, 0.863, 0.866, 0.867, 0.874 |
| | with defense | 0.627, 0.631, 0.632, 0.632, 0.633, 0.634, 0.643, 0.644, 0.661, 0.663 |

(b) Purchase

| Atttack methods | with/wo defense | Top-10 Attack AUC |
|---|---|---|
| Loss | wo defense | 0.699, 0.709, 0.715, 0.719, 0.727, 0.731, 0.735, 0.738, 0.763, 0.875 |
| | with defense | 0.663, 0.666, 0.671, 0.672, 0.684, 0.692, 0.694, 0.701, 0.705, 0.722 |
| Entropy | wo defense | 0.692, 0.697, 0.714, 0.714, 0.718, 0.724, 0.725, 0.725, 0.747, 0.868 |
| | with defense | 0.651, 0.651, 0.654, 0.657, 0.658, 0.673, 0.678, 0.681, 0.695, 0.711 |
| M-Entropy | wo defense | 0.699, 0.711, 0.716, 0.720, 0.727, 0.731, 0.734, 0.739, 0.765, 0.875 |
| | with defense | 0.663, 0.666, 0.671, 0.672, 0.684, 0.693, 0.694, 0.701, 0.705, 0.721 |
| Grad-x l2 | wo defense | 0.708, 0.725, 0.730, 0.733, 0.736, 0.738, 0.742, 0.747, 0.777, 0.897 |
| | with defense | 0.653, 0.662, 0.662, 0.667, 0.667, 0.669, 0.672, 0.684, 0.696, 0.697 |
| Grad-w l2 | wo defense | 0.662, 0.664, 0.665, 0.666, 0.666, 0.668, 0.670, 0.671, 0.681, 0.710 |
| | with defense | 0.629, 0.629, 0.632, 0.634, 0.634, 0.638, 0.639, 0.654, 0.655, 0.663 |

**Table 7:** Top-10 Attack AUC among 100 label classes on (a) Texas and (b) Purchase with and without (wo) applying our defense. The AUC values are shown in increasing order.

## C.9 PRIVACY-UTILITY CURVES

In this section, we include detailed results with regard to various datasets and different target models' architectures: we show results on CIFAR-10 dataset with VGG11 architecture in Figure 9; CIFAR-100 dataset with VGG11 architecture in Figure 10; CH-MNIST with ResNet20 architecture in Figure 11; Texas100 with MLP architecutre in Figure 12; Purchase100 with MLP architecutre in Figure 13.

We observe that while baseline approaches are effective for at most one data modality, our approach is the only one that is consistently effective in defending MIAs, across all different datasets and model architectures.

**Natural Images.** See Figure 3(a)-3(b) in main paper, and Figure 9-10: for natural image datasets (CIFAR-10 and CIFAR-100), DP-SGD is the best baseline method in terms of defending MIAs and preserving model utility, but is inferior to RelaxLoss as our method consistently achieve better test accuracy (model utility) across the full range of achievable privacy level.

Among all regularization-based defense methods, Early-stopping is the only one that exhibits noticeable effects in reducing attack AUCs. In comparison, our approach can achieve the same level of defense effectiveness with a much better model utility. Moreover, our approach can further decrease the attack AUC to a random-guessing level, which is not achievable by Early-stopping.

Memguard and Adv-Reg, previous state-of-the-art defense mechanisms specifically designed for MIAs, are highly effective in defending NN-based attack but generally lose their effectiveness for other types of attacks. In comparison, our approach shows much better defense effectiveness for all types of attacks while achieving better model utility at the same time.

**Gray-scale Medical Images** See Figure 11: for gray-scale medical image (CH-MNIST), our approach is comparable with the best baseline methods (i.e., Confidence-penalty), as both approaches are able to reduce the MIAs to a random-guessing level while preserving the model utility.

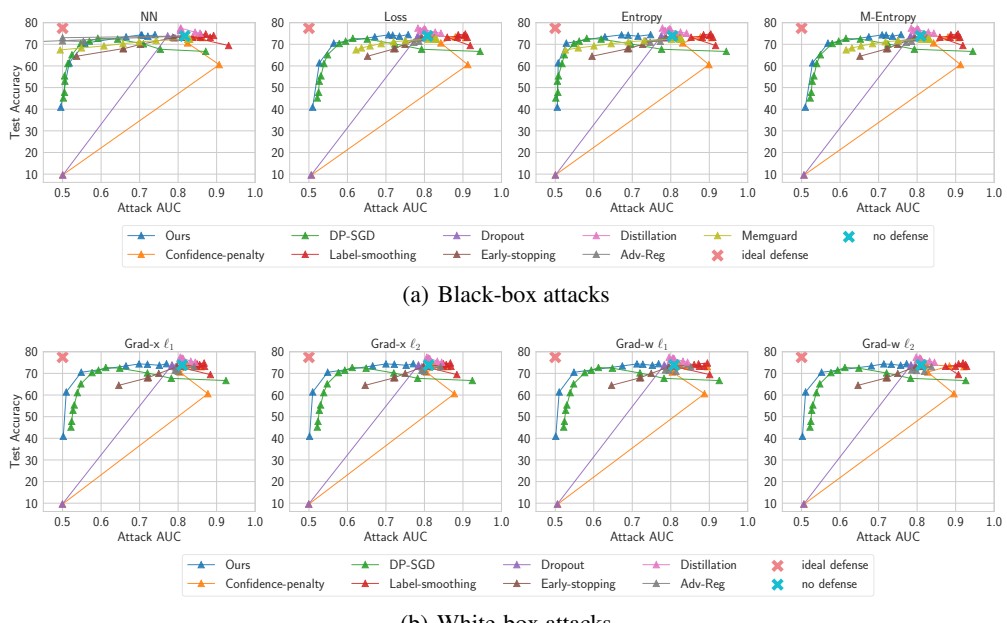

**Figure 9:** Comparisons of all defense mechanisms on CIFAR-10 dataset (VGG11). We set the clipping bound $C = 0.5$ and vary the noise scale over $0.01\text{-}0.45$ for DP-SGD. We vary the noise magnitude across $0\text{-}20$ for Memguard.

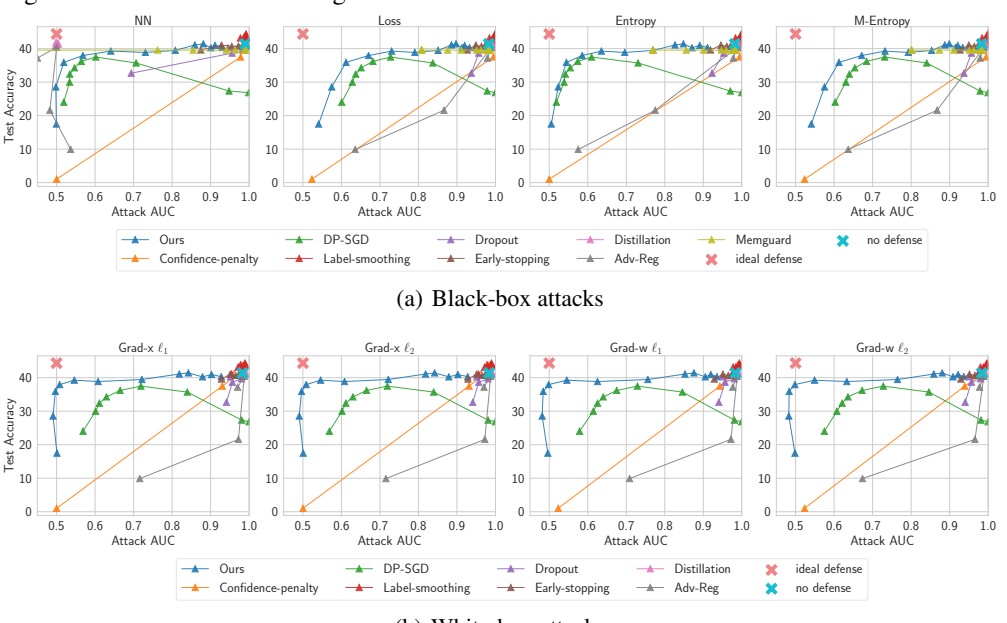

**Figure 10:** Comparisons of all defense mechanisms on CIFAR-100 dataset (VGG11). We set the clipping bound $C=1.0$ and vary the noise scale over $0.01\text{-}0.3$ for DP-SGD. We vary the noise magnitude across $0\text{-}400$ for Memguard.

In comparision, DP-SGD is significantly worse than our approach for this data modality, as it inevitably degrades the model utility.

Memguard and Adv-Reg are still highly effective in defending NN-based attack. Moreover, Memguard is able to defend black-box MIAs to a random-guessing level without degrading the model utility, but is not applicable to white-box attacks. In contrast, our approach is applicable to all attacks, and achieves comparable (or better) effectiveness for black-box attacks.

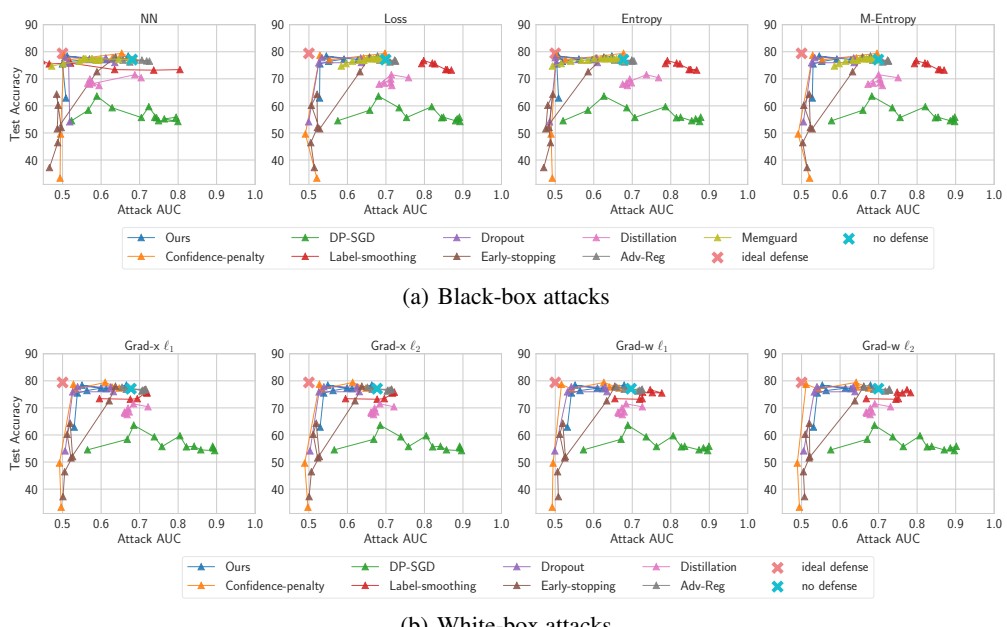

(a) Black-box attacks

(b) White-box attacks

**Figure 11:** Comparisons of all defense mechanisms on CH-MNIST dataset. We set the clipping bound $C$=5.0 and vary the noise scale over 0.001-0.5 for DP-SGD. We vary the noise magnitude across 0-50 for Memguard.

**Binary Medical and Transaction Records.** See Figure 12-13: for binary medical and transaction records (Texas100 and Purchase100), Label-smoothing performs generally the best among all baseline methods. Compared to our approach, Label-smoothing can achieve superior model utility when the Attack AUC is of range around 0.65-0.8 on Texas100 and 0.55-0.65 on Purchase100. However, our approach is able to reduce the attack AUC to around the random-guessing level, which is not always possible for Label-smoothing (white-box attacks on Purchase100, black-box attacks on both datasets).

DP-SGD exhibits unexpected results for these two datasets: small noise scale results in both lower Attack AUC and higher model utility, which is contradictory to the common belief that small-scale noise only provides weak privacy guarantee and thus the Attack AUC will remain high. We conjecture that there exists a non-negligible gap between the worst-case privacy guarantee that provided by the theoretical privacy analysis and the real-world attack performance in practice. Especially for the small-scale noise case, the privacy cost $\epsilon$ is meaninglessly large and cannot faithfully reflect the risk when facing practical MIAs. We tried varying the noise scale in the experiments: by decreasing the noise scale, we find DP-SGD cannot reduce the Attack AUC to random-guessing level, while by increasing the noise scale, the utility soon drops significantly. In comparison, our approach consistently yields better privacy-utility trade-off.

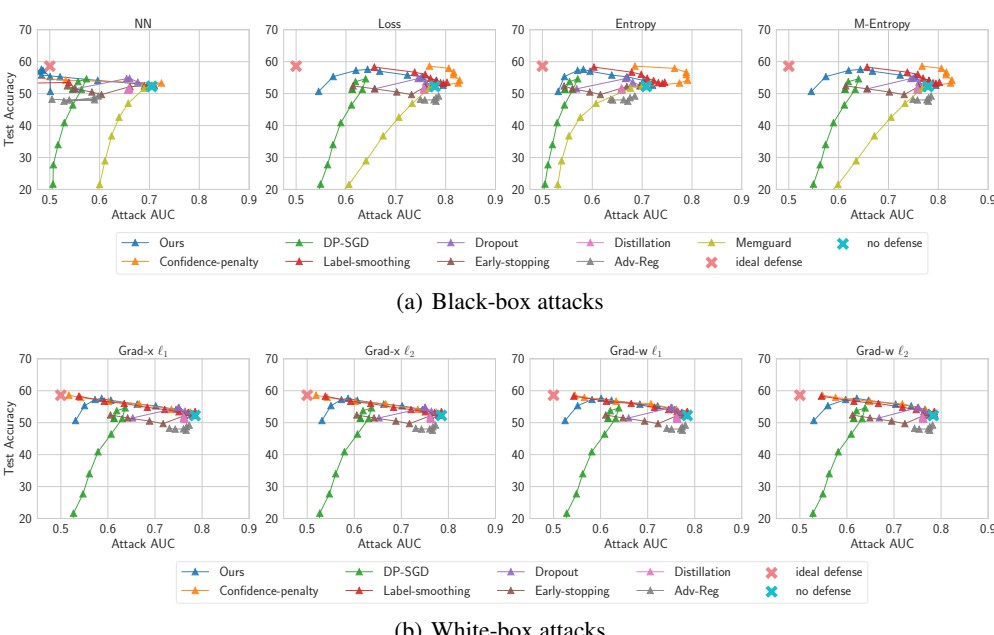

(a) Black-box attacks

(b) White-box attacks

**Figure 12:** Comparisons of all defense mechanisms on Texas100 dataset. We set the clipping bound $C$=1.0 and vary the noise scale over 0.001-0.5 for DP-SGD. We vary the noise magnitude across 0-500 for Memguard.

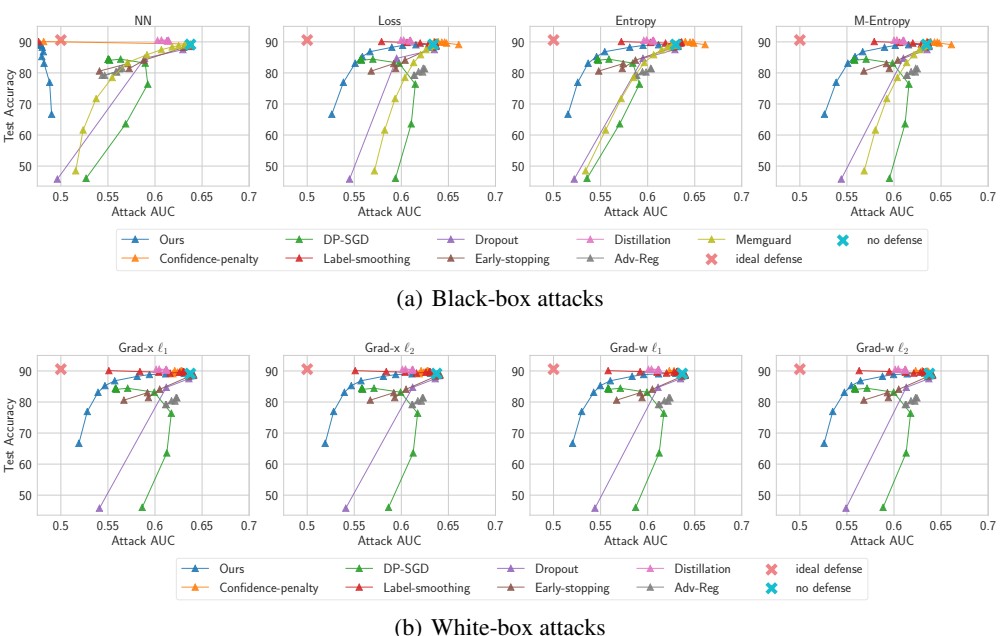

(a) Black-box attacks

(b) White-box attacks

**Figure 13:** Comparisons of all defense mechanisms on Purchase100 dataset. We set the clipping bound $C$=1.0 and vary the noise scale over $10^{-4}$-0.4 for DP-SGD. We vary the noise magnitude across 0-300 for Memguard.

## C.10 LOSS HISTOGRAMS

To better understand the effect of each defense method, we additionally plot the loss histograms when applying different defense methods on target models with a ResNet20 architecture trained on CIFAR-10 dataset in Figure 14-20. In the parentheses of each subtitle, we show the hyper-parameter values corresponding to each subfigure from left to right.

We observe that: *(i)* Regularization techniques in general have limited effects in reducing the gap between the training and testing loss distributions. *(ii)* Unlike our approach (See Figure 1 in the main paper), baseline methods are generally not able to increase the training loss variance nor closing the gaps between the member and non-member distributions, which explains the superior performance of our approach in defending various types of MIAs. *(iii)* By setting a relatively large noise scale, DP-SGD is able to increase the training loss variance and reducing the gap between the training and testing loss values (See last column of Figure 17). However, the large scale of noise dampen the learning signal in this case, leading to a non-negligible utility drop.

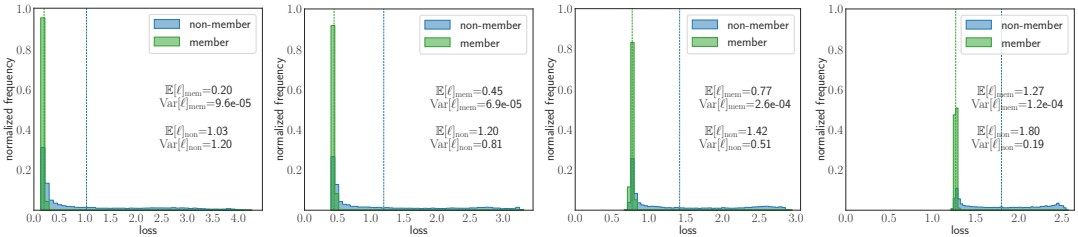

**Figure 14:** Loss histograms when applying Label-smoothing ($\alpha = 0.2, 0.4, 0.6, 0.8$).

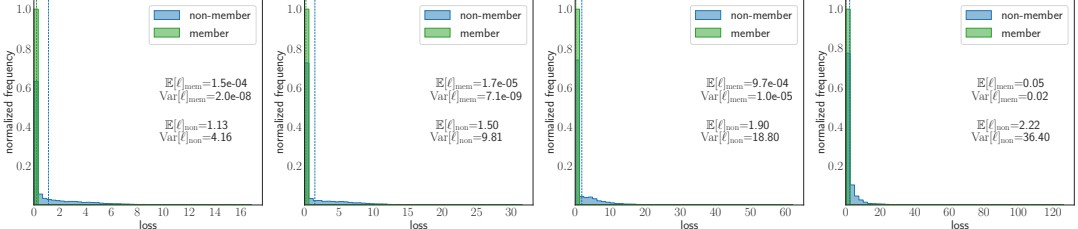

**Figure 15:** Loss histograms when applying Dropout (dropout rate = 0.1, 0.5, 0.7, 0.9)

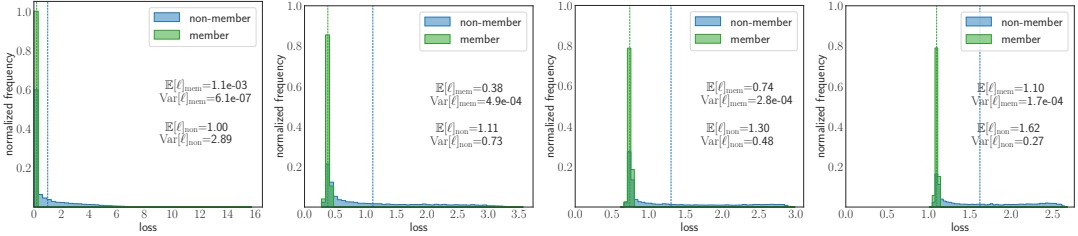

**Figure 16:** Loss histograms when applying Confidence-penalty ($\alpha = 0.1, 0.5, 1.0, 2.0$)

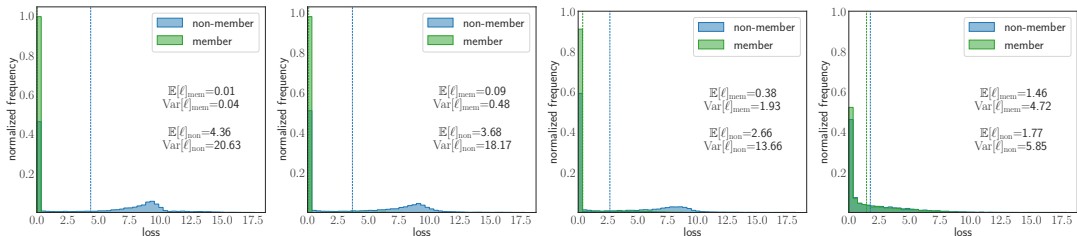

**Figure 17:** Loss histograms when applying DP-SGD (noise scale = 0.01,0.05,0.1,0.5)

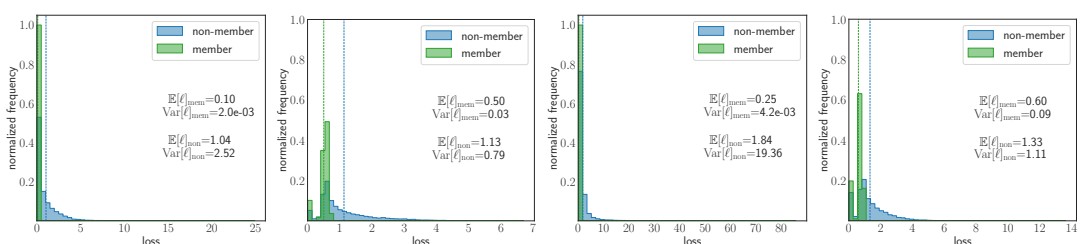

**Figure 18:** Loss histograms when applying Adv-Reg ($\alpha$ = 0.8,1.0,1.2,1.4)

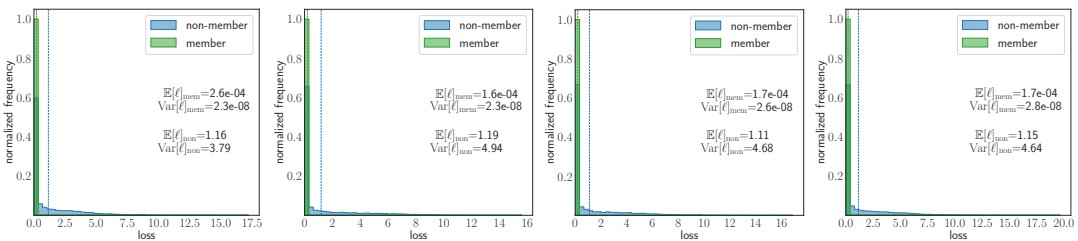

**Figure 19:** Loss histograms when applying Distillation ($T$ = 1,10,50,100)

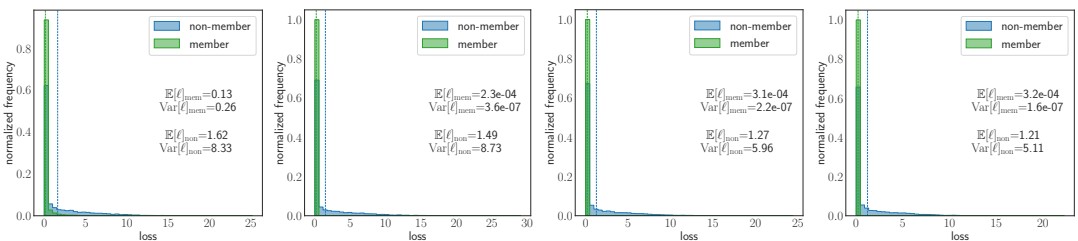

**Figure 20:** Loss histograms when applying Distillation Early-stopping (ep = 25,50,75,100).

## C.11 ANALYSIS OF MODEL GENERALIZATION

As supplementary to Section 7 of our main paper, we show results of toy experiments that investigate the impact of our approach on model generalization. We visualize the prediction scores and the decision boundaries in Figure 21. In contrast to vanilla training that assigns high confidence scores on hard examples near the decision boundary, our approach can soften the decision boundaries, leading to a large area with low (and well-calibrated) predicted confidence scores. In line with Zhang et al. (2017); Pereyra et al. (2017), we conjecture that the flatness of decision boundaries improves model generalization, while an in-depth analysis is left as future work.

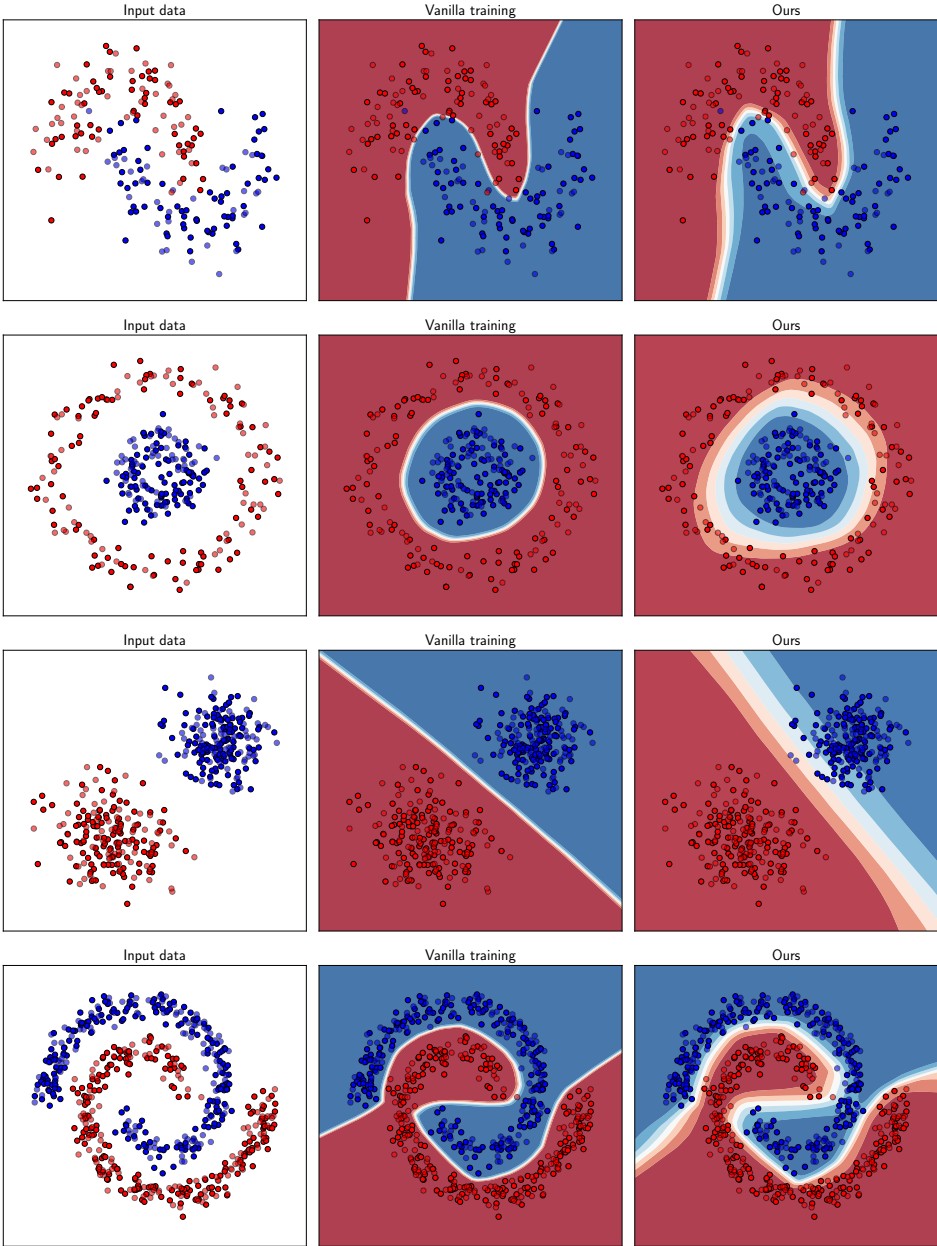

**Figure 21:** Visualization of target models' prediction scores and decision boundaries. The training samples are shown in solid colors and testing points are semi-transparent.

