# OpenReview forum: "RelaxLoss: Defending Membership Inference Attacks without Losing Utility"
_ICLR.cc/2022/Conference — ICLR 2022 Spotlight_

### Official Review · Reviewer_sYYW · 2021-11-02

**Correctness:** 3
**Technical Novelty And Significance:** 3
**Empirical Novelty And Significance:** 3
**Recommendation:** 8
**Confidence:** 4

**Main Review:**

[strengths]
1. The two techniques – relaxed target loss with gradient ascent and confidence flattening are well-motivated in Section 4 and 5.
2. The paper thoroughly evaluates the defense method on 5 datasets, and compares it with 8 defense baselines, making the experimental results very convincing.
3. The consideration of adaptive attacks in Section 6.4 is necessary and important.

[weaknesses]
1. Section 6.4 only considers adaptive attacks for NN attack, can you run adaptive attacks for other attack methods as well and report the best attack success among all methods in Table 2 (or put results in another new table)?
2. A small presentation issue: subfigures in Figure 3 are too small. Maybe put some of them in Appendix and enlarge the main subfigures?


**Summary Of The Paper:**

The paper proposes a new training algorithm to defend against membership inference attacks (MIA) in machine learning models. Motivated by the connection between MIA success and difference between training and test loss distributions, the proposed algorithm sets a positive target mean training loss value and applies gradient ascent if the average loss of current training batch is smaller than it. Furthermore, to avoid hurting model accuracy, the proposed algorithm also flattens the probabilities among incorrect labels during training steps. Extensive results on multiple datasets along with several defense baselines validate the effectiveness of the proposed defense idea.

**Summary Of The Review:**

Overall, I like this paper. The used techniques are well-motivated. Also, the experimental evaluation is very thorough to include several datasets, multiple existing defense baselines, and include adaptive attacks. So, I recommend accepting this paper.

---

> ### Author Response · Authors · 2021-11-18
> **Response to Reviewer sYYW**
>
> We thank all reviewers for their valuable feedback. We are encouraged that all reviewers voted to accept, finding our work **"well-motivated"** and **"clear"**, the proposed method is **"effective"**, **"computationally efficient"**, and **"close to optimal in maintaining the trade-off between the utility and attack performance"**, the experimental results are **"extensive"** and **"very convincing"**, and that both the technical and empirical contributions of the paper were judged to be **"significant"**.
>
> We now address individual concerns of **Reviewer sYYW**. All minor points are incorporated in the revised manuscript, where the modified text is marked in **blue**.
>
>
> **1. [Section 6.4 only considers adaptive attacks for NN attack, can you run adaptive attacks for other attack methods as well and report the best attack success among all methods in Table 2 (or put results in another new table)?]**
> Thanks for the suggestion. We include the results of adaptive attacks for all different attack methods in Appendix C.4 of the revised manuscript, and report the *highest* adaptive attack accuracy in Table 2. We observe that our approach *consistently* reduces the accuracy of all types of adaptive attacks (4.9%-47.1% relative difference) compared to vanilla training.
>
>
> **2. [A small presentation issue: subfigures in Figure 3 are too small. Maybe put some of them in Appendix and enlarge the main subfigures?]**
> Thanks for the suggestion.  As each subfigure in Figure 3 corresponds to different attack methods with distinct properties, we believe it is more convincing to include all of them and place them side by side in the main paper. Also, we have adjusted the figure size and font size s.t. the curves, captions, and text in Figure 3 are readable.

---

### Official Review · Reviewer_zx7i · 2021-11-02

**Correctness:** 3
**Technical Novelty And Significance:** 3
**Empirical Novelty And Significance:** Not applicable
**Recommendation:** 8
**Confidence:** 4

**Main Review:**

Positive aspects:
-	The proposed method outperforms similar defense strategies while preserving the accuracy of the classification model on test data.
-	Even with the full knowledge of the attacker on the defense mechanism, the defense algorithm is capable of reducing the attacker’s accuracy considerably.
-	Their defense strategy is computationally efficient (in comparison with other similar methods) and blind to the attack algorithm.

Negative aspects:
-	The paper lacks intuition behind some assumptions and choices made in designing the algorithm (more below).
-	The authors did not discuss limitations of the method (more below).
-	Some minor errors that can be easily fixed (see Minor Comments).

---------------------------------------------------
Major Comments:
-	In the set of defender assumptions in Section 3, the necessity for considering the first assumption is not clear. In practice, additional unlabeled data is normally available and if they can help with building a stronger defense mechanism, why should we ignore them? This assumption lacks justification.
-	I find the theoretical and intuitive justification behind the “posterior flattening” step to be lacking. By discarding the learned information of the network about the non-ground-truth classes and manually replacing the values, my intuition is that there is a good chance we are overfitting (i.e., we can't expect good performance on the test set). The poor testing results in Table 1 for purchase100 data, which has the largest number of training samples, might be hinting at this overfitting. So, it seems the method has its own limitations with regards to the training sample size. I believe authors should discuss this limitation (and any other possible ones) and address them.
-	Figure 4 would be more informative if the comparison was made against the best performing baseline instead.
-----------------------------------------------------
Minor Comments:
-	In “Notations” (section 3), the sentences are confusing. There are no x, y , p in f(.;theta), and the notation for “1” is not consistent.
-	Similarly, in "Attacker’s Assumptions”, the definition of m and z should precede that of S.
-	If the attack has full access to the model (as mentioned in Attacker’s Assumptions), it means the attacker has full knowledge on the classifier’s architecture and parameters. So, shouldn’t it be A(z,f(.;theta)) instead of A(z,theta)?
-	Figure 1-c, the (E,Val) pairs should be linked to the corresponding distribution.


**Summary Of The Paper:**

The study tackles the problem of defense against membership inference attack, with a focus on (1) decreasing the performance of the attack, (2) maintaining the classifier’s performance, (3) assuming the blindness towards the attack model. They achieve (1) by closing the distance between the train and test distributions and maintain the utility of the model by flattening the posterior scores of the non-target classes. Their method is shown to be computationally efficient (i.e., the additional computational cost is negligible), close to optimal in maintaining the trade-off between the utility and attack performance, and effective in the face of attack’s countermeasures.

**Summary Of The Review:**

The approach is clear, and the extensive experiments performed on variety of datasets against reliable baselines proves the merit of the proposed defense algorithm. However, the intuition behind some assumptions and steps in the algorithm is not clear and authors should certainly discuss the limitations of their method.

---

> ### Author Response · Authors · 2021-11-18
> **Response to Reviewer zx7i**
>
> We thank all reviewers for their valuable feedback. We are encouraged that all reviewers voted to accept, finding our work **"well-motivated"** and **"clear"**, the proposed method is **"effective"**, **"computationally efficient"**, and **"close to optimal in maintaining the trade-off between the utility and attack performance"**, the experimental results are **"extensive"** and **"very convincing"**, and that both the technical and empirical contributions of the paper were judged to be **"significant"**.
>
> We now address individual concerns of **Reviewer zx7i**. All minor points are incorporated in the revised manuscript, where the modified text is marked in **blue**.
>
> **1. [Section 3: the necessity for considering the first assumption is not clear. In practice, additional unlabeled data is normally available and if they can help with building a stronger defense mechanism, why should we ignore them?]**
> To the best of our knowledge, the usage of additional public data is not standard and mostly requires dedicated domain adaptation techniques (or assumes that both the public and the private data follow the same distribution, which is normally not the case). Especially for non-image datasets (e.g. Texas100 and Purchase100 in our submission), we are not aware of any existing methods that exploit public sources of additional data. We agree that the investigation of the benefits brought from additional data is a highly interesting topic, but is orthogonal to the main contribution of our work.
>
> **2. [Justification behind the “posterior flattening” step is lacking. By discarding the learned information of the network about the non-ground-truth classes and manually replacing the values, my intuition is that there is a good chance we are overfitting (i.e., we can't expect good performance on the test set). The poor testing results in Table 1 for purchase100 data, which has the largest number of training samples, might be hinting at this overfitting. So, it seems the method has its own limitations with regards to the training sample size.]**
> Our insight is that the “posterior flattening” step encourages a certain margin between the ground-truth class and the most confusing non-ground-truth class, and its effectiveness is justified in the ablation study (Section 6.5). Though the posterior flattering step encourages a uniform prediction score for all non-ground-truth classes, we believe that the learned information about the non-ground-truth classes is not discarded: e.g., if non-uniformity among non-ground-truth classes is preferable for the learning task, it will be kept or even amplified in the following gradient descent/ascent steps. The improvement of our approach in both the top-1 and top-5 (sometimes even larger improvement than top-1) accuracy should help to support this point. Our intuition is that our approach lets the model learn to balance the ground-truth and all non-ground-truth class prediction scores s.t. it is able to make correct predictions while suppressing the signals that can be exploited by MIAs.
>
> For the Purchase100 data, we find that none of the baseline methods can effectively improve the test accuracy. We attribute this to a saturated performance achieved by vanilla training, while a more in-depth interpretation of this phenomenon requires understanding the model generalization, which is out of the scope of our work. However, we do not see a particular connection to our “posterior flattening” step.
>
> **3. [I believe authors should discuss this limitation (and any other possible ones) and address them]**
> Thanks for the suggestion. We discuss the possible limitations in Appendix C.1 of the revised manuscript.
>
> **4. [Figure 4 would be more informative if the comparison was made against the best performing baseline instead.]**
> Thanks for the suggestion. However, it is generally hard to identify the best performing baseline method for Figure 4, as baseline methods either degrade the model utility (our method in Figure 4 is *without* utility loss) or are not effective in defending MIAs. Thus, we decide to show the comparison with baselines via the privacy-utility curves in Figure 3 and Appendix Figure 9-13.
>
> **5. [Minor comments: Notations in Section 3 and Figure1. ]**
> Thanks for the suggestions. We have modified the Notations section (First paragraph of Section 3) and adjusted the figure.

---

> > ### Comment · Reviewer_zx7i · 2021-11-22
> > **Post-Rebuttal Update**
> >
> > I appreciate the authors' response to all reviewers. I believe they have answered both my and other reviewers' questions and concerns fairly. So, I am willing to keep my positive evaluation of the paper and vote for accepting the paper.

---

### Official Review · Reviewer_A7eb · 2021-11-04

**Correctness:** 3
**Technical Novelty And Significance:** 3
**Empirical Novelty And Significance:** 3
**Recommendation:** 8
**Confidence:** 2

**Main Review:**

Strengths:
This paper proposed a relax loss to defend privacy leakage, since the authors find that membership privacy risks can be reduced by narrowing the gap between the loss distributions. Also the gradient descent and gradient ascent are used alternatively to balance the member and non-member loss distributions.

Weakness:
1. The authors give an analysis for "RelaxLoss increases the variance of the training loss distribution" in appendix A. In Theorem A.1, the authors show that the variance of loss distribution increases after a gradient ascent step, with the condition $\text{Cov}(l, |\Delta l|)>0$. My questions here are why we need the absolute operator. The variance of loss distribution after a gradient ascent step should be $\text{Var} (l+\Delta l)$.  In this case, $\Delta l$ can be negative, and how to make sure the condition $\text{Cov}(l, |\Delta l|)>0$ still holds. Also, the assumption of "each sample within a batch exhibits the same gradient" seems too strong and impractical. The common assumptions may be: each sample within a batch exhibits the same norm of gradient.

2. In Figure 5, why does 'w/o gradient ascent' only have a single point? It should be a curve just like 'ours' and 'w/o posterior flattening'.

#############after authors' response#############
Thanks for the responses from the authors. I will raise my score to 8.


**Summary Of The Paper:**

To defend the common membership inference attacks, this paper proposes a relaxed loss with a more achievable learning target to reduce the attack accuracy and also help with the generalization gap of learning models.  Extensive evaluations on five datasets with diverse modalities show that the proposed method can achieve higher membership inference protection.

**Summary Of The Review:**

This paper proposes a relaxed loss to narrow the loss gap and reduce the distinguishability between the training and testing loss distributions, and further prevent the privacy leakage. However, I have some questions about the theoretical analysis, i.e., why the "RelaxLoss increases the variance of the training loss distribution".

---

> ### Author Response · Authors · 2021-11-18
> **Response to Reviewer A7eb**
>
> We thank all reviewers for their valuable feedback. We are encouraged that all reviewers voted to accept, finding our work **"well-motivated"** and **"clear"**, the proposed method is **"effective"**, **"computationally efficient"**, and **"close to optimal in maintaining the trade-off between the utility and attack performance"**, the experimental results are **"extensive"** and **"very convincing"**, and that both the technical and empirical contributions of the paper were judged to be **"significant"**.
>
> We now address individual concerns of **Reviewer A7eb**. All minor points are incorporated in the revised manuscript, where the modified text is marked in **blue**.
>
>
> **1. [Why we need the absolute operator in Theorem A.1. The variance of loss distribution after a gradient ascent step should be Var(l+Δl). In this case,  Δl can be negative, and how to make sure the condition Cov(l,|Δl|)>0 still holds. ]**
> The absolute operator does not affect the analysis in Theorem A.1 (See our revised manuscript), and is required for the intuitive interpretation: we interpret $|\Delta \ell|$ as the rate of loss changes. More details can be found in the revised manuscript.
>
> **2. [The assumption of "each sample within a batch exhibits the same gradient" seems too strong and impractical. The common assumptions may be: each sample within a batch exhibits the same norm of gradient.]**
> Thanks for the suggestion. We can still derive the results in Corollary A.1 with a more realistic assumption: the gradients within a batch have the same norm and are well-aligned (instead of being exactly the same). More details can be found in Corollary A.1 in our revised manuscript. We admit that our current derivation does not analyze the stochasticity of the training process, while we believe that our analysis does shed light on the property of our proposed approach and we leave a more in-depth theoretical investigation as future work.
>
> **3. [In Figure 5, why does 'w/o gradient ascent' only have a single point? It should be a curve just like 'ours' and 'w/o posterior flattening'.]**
> For the ‘w/o gradient ascent’ configuration, the hyperparameter (i.e., the ‘relaxed’ target loss) that determines the privacy-utility trade-off no longer exists. As there is no more tunable hyperparameter that can control the privacy level, we only have a single point in Figure 5.

---

### Decision · Program_Chairs · 2022-01-20

**Decision:**

Accept (Spotlight)

**Comment:**

The paper proposes an approach and specific training algorithm to defend against membership inference attacks (MIA) in machine learning models. Existing MIA attacks are relatively simple and rely on the test loss distribution at the query point and therefore the proposed algorithm sets a positive target mean training loss value and applies gradient ascent if the average loss of current training batch is smaller than it (in addition to the standard gradient descent step). The submission gives extensive experimental results demonstrating advantage over existing defense methods on several benchmarks. The primary limitation of the work is that it defends only against rather naive existing attacks which do not examine the model (but rely only on the loss functions).